# DiMa: Understanding the Hardness of
# Online Matching Problems via Diffusion Models

**Boyu Zhang** [* 1 2 3]  **Aocheng Shen** [* 1]  **Bing Liu** [1]  **Qiankun Zhang** [✉ 1 2 3]  **Bin Yuan** [1 3 4 5]  **Jing Wang** [6]
**Shenghao Liu** [1]  **Xianjun Deng** [1]

## Abstract

We explore the potential of *AI-enhanced combinatorial optimization theory*, taking online bipartite matching (OBM) as a case study. In the theoretical study of OBM, the *hardness* corresponds to a performance *upper bound* of a specific online algorithm or any possible online algorithms. Typically, these upper bounds derive from challenging instances meticulously designed by theoretical computer scientists. Zhang et al. (ICML 2024) recently provide an example demonstrating how reinforcement learning techniques enhance the hardness result of a specific OBM model. Their attempt is inspiring but preliminary. It is unclear whether their methods can be applied to other OBM problems with similar breakthroughs. This paper takes a further step by introducing DiMa, a unified and novel framework that aims at understanding the hardness of OBM problems based on denoising diffusion probabilistic models (DDPMs). DiMa models the process of generating hard instances as denoising steps, and optimizes them by a novel reinforcement learning algorithm, named *shortcut policy gradient* (SPG). We first examine DiMa on the classic OBM problem by reproducing its known hardest input instance in literature. Further, we apply DiMa to two well-known variants of OBM, for which the exact hardness remains an open problem, and we successfully improve their theoretical state-of-the-art upper bounds.

---

[*]Equal contribution  [1]School of Cyber Science and Engineering, Huazhong University of Science and Technology, Wuhan, China [2]Key Laboratory of Cyberspace Security, Ministry of Education, Zhengzhou, China [3]Hubei Key Laboratory of Distributed System Security, Wuhan, China [4]Songshan Laboratory, Zhengzhou, China [5]Visiting researcher with the Lion Rock Labs of Cyberspace Security, CTlHE, Hong Kong, China [6]School of Software Engineering, Huazhong University of Science and Technology, Wuhan, China. Correspondence to: Qiankun Zhang <qiankun@hust.edu.cn>.

*Proceedings of the 42^{nd} International Conference on Machine Learning*, Vancouver, Canada. PMLR 267, 2025. Copyright 2025 by the author(s).

## 1. Introduction

Online bipartite matching (OBM, Karp et al. (1990)) is one of the most fundamental and central problems in online optimization, due to its broad applications in economics, operations research, and computer science. It captures varies practical domains such as Internet advertising(Mehta et al., 2007; Huang et al., 2020; 2023), resource allocation(Braun et al., 2024; Zhang et al., 2024b; Ekbatani et al., 2023), and transportation(Haliem et al., 2020; Dutta & Sholley, 2018; Asghari et al., 2016). Generally, OBM studies how to match one side of *items* arriving one by one to the other side of *buyers*, who are known upfront. The optimization objective is to maximize the number of matches found by an *online algorithm*, which can only make immediate and irrevocable decisions sequentially. In the theoretical study of OBM, a standard metric for measuring the performance of an online algorithm is called *competitive ratio* (CR), which is defined as the ratio between the matching size found by the algorithm and the *optimal offline* matching on *any* input bipartite graph instances. CRs range from $0$ to $1$ due to the uncertainty of the online nature, and the closer they are to $1$, the better they are. Extensive works study OBM and related problems by improving CRs either theoretically or empirically. We refer readers to see (Devanur & Mehta, 2022; Mehta et al., 2013) for surveys.

Our work lies at *AI-enhanced combinatorial optimization theory*. A recent work by Zhang et al. (2024a) gives the first successful attempt. They focus on the *upper bound* (also known as the hardness result) of the CR, which is a theoretical bound of either *an online algorithm* or *a problem*. In online optimization theory, the upper bound of an algorithm corresponds to the construction of a family of *hard instances* for that algorithm, whereas the upper bound of a problem corresponds to instances proved to be hard for *all* algorithms. An improved upper bound often emerges from a smarter construction of hard instances, which can be highly non-trivial and heavily rely on the expertise of a theoretical computer scientist. Zhang et al. (2024a) improves the best-known upper bound of an open-ended OBM problem, by collaborating with the reinforcement learning (RL) approach.

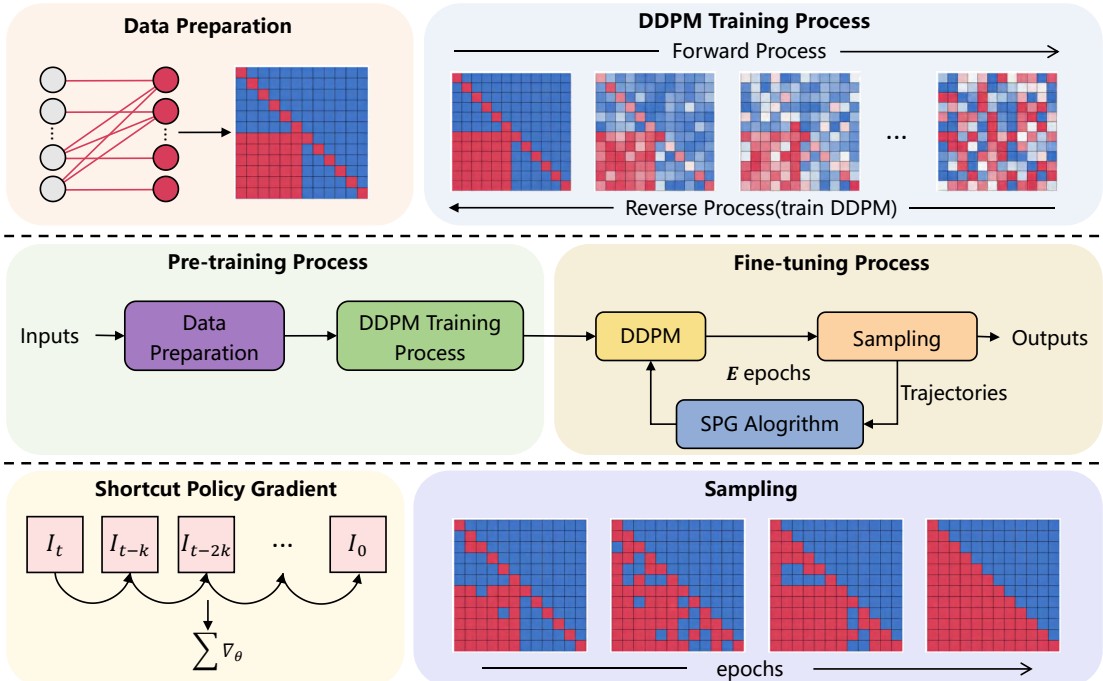

*Figure 1.* Overview of DiMa framework.

Their attempt, while interesting and successful in improving the theoretical state-of-the-art (SOTA), is yet preliminary. First, they model the action of the Markov decision process (MDP) as determining an online vertex at each time step, resulting in an action space of exponential size. It leads to poor generalizability to generate large-scale input instances and destroys the complete information of the graph structure. Furthermore, even though their approach is fundamentally end-to-end, the training process still requires expert insights to identify learned patterns in instances, and continually modify the model's action space. As a consequence, it remains unclear whether their techniques can be applied to a broader range of other OBM problems with improved bounds on the harness results. Finally, their model is essentially training from scratch and thus does not effectively utilize the distribution of hard instances known in the literature. This paper delves into exploring the potential of AI-enhanced OBM theory by introducing a unified and novel framework, named *Diffusion for Matching* (DiMa), that generates hard instances for OBM problems based on a denoising diffusion probabilistic model (DDPM) (Ho et al., 2020).

DDPMs are known to be powerful in novel image generations while preserving underlying distributions of known samples. However, applying them to generate hard instances for OBM can be non-trivial. Merely sampling new input instances from a learned distribution of known hard-case instances may not be enough to generate worse ones, and

therefore, obtain an improved upper bound. To address it, our DiMa follows a pre-train-then-fine-tune paradigm. During pre-training, DiMa trains an image-based DDPM to generate instances of a distribution that approximates the known hard instance distribution. In fine-tuning, DiMa models the denoising step as MDP and formulates it as policy optimization. To make the optimization robust and efficient, we propose a novel *shortcut policy gradient* (SPG) algorithm tailored for DiMa. DiMa finally produces harder instances for a bunch of OBM problems. Figure 1 presents an overview of DiMa.

Our contributions are concluded as follows:

- We propose DiMa, the first DDPM-based framework that generates novel hard instances for OBM problems.

- We examine DiMa on the classic (fractional) OBM model, whose hardest instance is known to be a *upper-triangular* distribution. Our DiMa succeeds in converging at such distribution.

- We apply DiMa to two open-ended OBM problems, whose worst-case instances remain unknown, including *online matching with random arrivals* (Goel & Mehta, 2008) and *online matching with stochastic rewards* (Mehta & Panigrahi, 2012). We improve their SOTA theoretical upper bounds of CRs to $0.723$ and $0.594$, respectively, based on the instances generated by DiMa.

## 2. Related Works

This section discusses only previous works most relevant to us. The rest is referred to Appendix A.

**ML for OBM.** Little is known other than Zhang et al. (2024a) in the category of AI-enhanced combinatorial optimization theory, which may be inspired by the recent success in ML methods for mathematical problems (Trinh et al., 2024; Romera-Paredes et al., 2024; Wang & Deng, 2020; Wu et al., 2020; Huang et al., 2024; Lin et al., 2024). However, there are a series of works (Kong et al., 2018; Du et al., 2021; Zuzic et al., 2020) reveals that ML models can be trained to approximate the behavior of the known theoretical optimal online algorithm. Du et al. (2021) and (Zuzic et al., 2020) also report that an adversarially trained model can reproduce the known worst instances in the online knapsack problem and the AdWords problem, respectively. It is unclear whether their methods can be used to explore completely new instances that do not appear in the literature. There are other works aiming at learning empirically efficient algorithms for various OBM problems (Alomrani et al., 2021; Li et al., 2023; Hayderi et al., 2024; Xie et al., 2023).

**Optimizing DDPMs by RL.** On the technical aspect, our method benefits from several recent works on optimizing diffusion models to align with downstream objectives. DPOK (Fan et al., 2024) and DDPO (Black et al., 2023) align text-to-image DDPMs with black-box reward signals; AlignProp (Prabhudesai et al., 2023) and DRaFT (Clark et al., 2023) optimize DDPMs with direct backpropagation for differentiable rewards; GDPO (Liu et al., 2024) aligns graph DPMs for arbitrary objectives. Our DiMa is inspired by their ideas of fine-tuning DDPMs using the RL approach, but novel in implementation details. Applying such an idea in the task of exploring novel hardness results of online algorithms can be highly non-trivial. Moreover, our SPG outperforms remarkably existing techniques in achieving a remarkably better performance-efficiency trade-off.

## 3. Preliminaries

We introduce the necessary backgrounds to understand DiMa in this section, including OBM, DDPMs, and RL.

### 3.1. Online Bipartite Matching

**Model.** An input instance $\mathcal{I}$ of a classic OBM model involves a bipartite graph $G = (L \cup R, \mathbb{E})$, with one side of *offline* vertices $L$ (buyers) known before the algorithm and another side of *online* vertices $R$ (items) arriving over time. When a vertex $r \in R$ arrives, the adjacent edges of $r$ are released, and the algorithm is asked to make an immediate and irrevocable decision to match $r$ to one of its unmatched neighbors in $L$. The objective is to maximize the matching size found by the algorithm.

**Competitive ratio.** The performance of an online algorithm, denoted as ALG, is generally compared to an *offline optimum*, denoted as OPT. An offline benchmark for the above OBM instance $\mathcal{I}$ is a standard (offline) bipartite matching problem on $\mathcal{I}$, where $L$, $R$, and $E$ are all given before a matching algorithm proceeds. *CR* is defined as $\inf_{\mathcal{I}} \frac{\text{ALG}(\mathcal{I})}{\text{OPT}(\mathcal{I})} \in [0, 1)$, meaning the infimum of ratio between ALG and OPT among all possible input instances.

**Hardness.** The hardness, which corresponds to a theoretical upper bound of CR, is either of *an algorithm* or *a problem* in online optimization theory. The algorithm hardness is typically derived from a family of hard instances, where the algorithm performs the worst. And the problem hardness is proved by constructing instances where *any* algorithms perform very badly. To achieve an improved upper bound of CR, a key ingredient is a better construction of input instances, which are harder for a specific algorithm or all possible algorithms.

**A warm-up example.** Consider the OBM model defined above. A simple Greedy algorithm matches any $r \in R$ to the *largest indexed* unmatched neighbor. It's not difficult to verify that Greedy is at least $0.5$-competitive. The hardness of Greedy is also known as $0.5$, which can be proved by a simple *z-graph* instance as follows. Suppose $L = \{\ell_1, \ell_2\}$ and $R = \{r_1, r_2\}$. $r_1$ arrives first and it is connected to both $\ell_1$ and $\ell_2$. $r_2$ arrives later and it has only one neighbor $\ell_2$. Note that Greedy will match $r_1$ to $\ell_2$ such that $r_2$ has no available neighbor, and thus ALG = 1. But the offline OPT can be 2 by matching $r_1$ to $\ell_1$ and $r_2$ to $\ell_2$. The hardness of the OBM problem is known to be $1 - \frac{1}{e} \approx 0.632$ by constructing an *upper-triangular* graph (Karp et al., 1990), which will be presented soon.

### 3.2. Denoising Diffusion Probabilistic Models

We adopt the standard DDPM for the instance generation. Formally, given an initialized instance sampled from a certain distribution $\mathcal{I}_0 \sim q(\mathcal{I})$, the DDPM defines a forward diffusion process which forms a Markov chain by adding Gaussian noise to $\mathcal{I}_0$ over $T$ steps, creating a sequence of instances $\mathcal{I}_1, \ldots, \mathcal{I}_T$ whose joint distributions are $q(\mathcal{I}_{1:T} | \mathcal{I}_0) = \prod_{t=1}^{T} q(\mathcal{I}_t | \mathcal{I}_{t-1})$. As $t$ increases, $\mathcal{I}_0$ loses its features, and $\mathcal{I}_T$ approaches an Gaussian distribution as $T$ tends to infinity. Reversing the diffusion process, which is also a Markov chain, allows to generate samples from $q(\mathcal{I}_{t-1} | \mathcal{I}_t)$ starting from Gaussian noise $\mathcal{I}_T \sim \mathcal{N}(0, I)$. The DDPM trains a model $p_\theta$ defined by $p_\theta(\mathcal{I}_0) = \int p_\theta(\mathcal{I}_{0:T}) \, d\mathcal{I}_{1:T}$ to approximate $q$, where $p_\theta(\mathcal{I}_{0:T}) = p(\mathcal{I}_T) \prod_{t=1}^{T} p_\theta(\mathcal{I}_{t-1} | \mathcal{I}_t)$. Training a DDPM is performed by optimizing a standard variational bound on negative log-likelihood $\mathbb{E}_q[-\log p_\theta(\mathcal{I}_0)]$. The objective can

be further written as:

$$\mathbb{E}_q \Bigg[ -\log p_\theta(\mathcal{I}_0|\mathcal{I}_1)$$

$$+ \sum_{t=2}^{T} D_{KL}\left[q(\mathcal{I}_{t-1}|\mathcal{I}_t,\mathcal{I}_0)\|p_\theta(\mathcal{I}_{t-1}|\mathcal{I}_t)\right] \Bigg], \quad (1)$$

where $D_{KL}[\cdot||\cdot]$ represents the KL divergence between two distributions.

### 3.3. Denoising Diffusion Implicit Model

Denoising diffusion implicit model (DDIM) (Song et al., 2020) achieves high-quality samples with significantly fewer steps compared to DDPMs. In DDPM, the reverse denoising process is explicitly probabilistic, meaning each timestep is conditioned on the noise added during the forward process. In contrast, DDIM uses an implicit sampling process, where the reverse process is parameterized such that no explicit noise prediction is needed at each step. Instead of iteratively denoising through probabilistic distributions, DDIM achieves faster sampling by skipping multiple steps in the reverse diffusion process while maintaining reasonable sample quality. Formally, DDIM samples a subsequence of length $S$, denoted as $[\tau_1, \ldots, \tau_S]$, from the original sequence $[1, \ldots, T]$. It defines $q(\mathcal{I}_{\tau_i}|\mathcal{I}_0) = \mathcal{N}(\mathcal{I}_{\tau_i}; \sqrt{\alpha_{\tau_i}}\mathcal{I}_0, (1-\alpha_{\tau_i})\mathbf{I})$, where $\alpha$ is a noise schedule parameter. By doing so, the reverse process becomes:

$$\mathcal{I}_{\tau_{i-1}} = \sqrt{\alpha_{\tau_{i-1}}}\left(\frac{\mathcal{I}_{\tau_i} - \sqrt{1-\alpha_{\tau_i}}\epsilon_\theta(\mathcal{I}_{\tau_i},\tau_i)}{\sqrt{\alpha_{\tau_i}}}\right)$$

$$+ \sqrt{1-\alpha_{\tau_{i-1}} - \sigma_{\tau_i}^2} \cdot \epsilon_\theta(\mathcal{I}_{\tau_i},\tau_i) + \sigma_{\tau_i}\epsilon, \quad (2)$$

where $\sigma_t = \eta \cdot \sqrt{(1-\alpha_{t-1})/(1-\alpha_t)} \cdot \sqrt{(1-\alpha_t)/\alpha_{t-1}}$ and $\epsilon \sim \mathcal{N}(0, I)$. Our SPG borrows DDIM's insight by skipping a few steps when computing the gradient in fine-tuning, achieving a remarkable performance-efficiency trade-off.

### 3.4. Markov Decision Process and Reinforcement Learning

An RL problem is typically formulated as a Markov Decision Process (MDP) (Feinberg & Shwartz, 2002) defined by $(\mathcal{S}, \mathcal{A}, p_0, P, R)$, where $\mathcal{S}$ represents the state space, $\mathcal{A}$ represents the action space, $P$ represents the transition function for state changes, $R: \mathcal{S} \times \mathcal{A} \to \mathbb{R}$ represents the reward function, and $p_0$ represents the initial state distribution. At each time step $t$, an agent perceives a state $s_t$ from the state space $\mathcal{S}$, executes an action $a_t$ from the action space $\mathcal{A}$, and gains a reward $R(s_t, a_t)$. Subsequently, it moves to a new state $s_{t+1}$, which is drawn from the transition probability distribution $P(s_{t+1}|s_t, a_t)$. The agent's behavior is guided by a policy $\pi$ that dictates the action $a$ to take given the state $s$. During its interactions with the MDP, the agent generates a trajectory $\tau$, which is a sequence comprising states,

actions, and rewards: $(s_0, a_0, r_0, ..., s_T, a_T, r_T)$. The RL model is trained to optimize the policy $\pi$ so as to maximize the agent's expected total reward gathered from trajectories generated by $\pi$, denoted as $\mathcal{J}_{\text{RL}}(\pi)$ and defined by:

$$\mathcal{J}_{\text{RL}}(\pi) = \mathbb{E}_{\tau \sim p(\tau|\pi)}\left[\sum_{t=0}^{T} r_t\right]. \quad (3)$$

## 4. Diffusion for Matching

This section details DiMa setup, including pre-training a DDPM and fine-tuning by RL. It also presents our *shortcut policy gradient algorithm* building on the classic policy gradient method REINFORCE (Mohamed et al., 2020).

### 4.1. Diffusion Model Pre-training

In our task of generating novel bipartite graph instances $G = (L \cup R, \mathbb{E})$, we without loss of generality assume $|L| = |R| = N$, and represent $G$ as an adjacency matrix instance $\mathcal{I} \in \{0, 1\}^{N \times N}$ to indicate the edges between online and offline vertices. Recall that in DDPM, we train a $p_\theta$ to approximate a known distribution $q$. In our context, we define $q$ as the known hard instances distribution for the OBM problem in the literature. For example, we may set $q$ as the z-graph instance as defined in the warm-up example of Section 3.1, involving some additional randomness. We preview that the construction of $q$ can be non-trivial, which we will discuss in our experiments. Given an initialized sample $\mathcal{I}_0 \sim q(\mathcal{I}_0)$, the forward process adds a small amount of Gaussian noise to the $\mathcal{I}_0$ in $T$ steps, producing a sequence of samples $\mathcal{I}_{1:T}$ with the distribution of:

$$q(\mathcal{I}_t \mid \mathcal{I}_{t-1}) = \mathcal{N}(\sqrt{1-\beta_t}\mathcal{I}_{t-1}, \beta_t I), \quad (4)$$

where $\beta_t$'s $\in (0, 1)$ are the noise schedule parameters. Running the reverse process to recover the instances requires a model $p_\theta$ to approximate $q$ as:

$$p_\theta(\mathcal{I}_{t-1} \mid \mathcal{I}_t) = \mathcal{N}(\mathcal{I}_{t-1}; \mu_\theta(\mathcal{I}_t, t), \Sigma_\theta(\mathcal{I}_t, t)), \quad (5)$$

where $\alpha_t = 1 - \beta_t$, $\bar{\alpha}_t = \prod_{i=1}^{t}\alpha_i$ and $\mu_\theta(\mathcal{I}_t, t) = \frac{1}{\sqrt{\alpha_t}}\left(\mathcal{I}_t - \frac{\beta_t}{\sqrt{1-\bar{\alpha}_t}}\epsilon_\theta(\mathcal{I}_t, t)\right)$. At this time, we predict the Gaussian noise added at each step through the model. The loss $L$ is defined as the difference between the reverse and forward noises: $L = \|\epsilon - \epsilon_\theta(\mathcal{I}_t, t)\|^2$. To ensure the differentiability during model training and instance sampling, all $\mathcal{I}_t$'s are viewed as real-number matrices. We convert them to 0-1-valued ones after the pre-training process.

### 4.2. DiMa Fine-tuning via RL

The pre-trained DDPM defines a sample distribution $p_\theta(\mathcal{I}_0)$, through its reverse denoising process $p_\theta(\mathcal{I}_{0:T})$. Given a competitive ratio calculator $\text{CR}(\cdot)$, the objective of the fine-tuning process is to minimize the expected CR over $p_\theta(\mathcal{I}_0)$:

$\mathcal{J}_{\text{CR}}(\theta) = \mathbb{E}_{\mathcal{I}_0 \sim p_\theta(\mathcal{I}_0)}[\text{CR}(\mathcal{I}_0)]$. However, directly optimizing $\mathcal{J}_{\text{CR}}(\theta)$ is challenging. Following DDPO (Fan et al., 2024), we formulate the reverse denoising process as a $T$-step MDP as:

$$s_t \triangleq (\mathcal{I}_{T-t}, T-t), \quad \pi(a_t \mid s_t) \triangleq p_\theta(\mathcal{I}_{T-t-1} \mid \mathcal{I}_{T-t}),$$

$$a_t \triangleq \mathcal{I}_{T-t-1}, \quad P(s_{t+1}|s_t, a_t) \triangleq (\delta_{\mathcal{I}_{T-t-1}}, \delta_{T-t-1}),$$

$$(s_t, a_t) \triangleq r(I_0) \quad \text{if } t = T, \quad R(s_t, a_t) \triangleq 0 \quad \text{if } t < T,$$

where the initial state $s_0$ is defined as the initial noisy instance $\mathcal{I}_T$, $\delta$ denotes the Dirac delta distribution and the policy aligns with the reverse transition probability distribution. As a consequence, the instance generation trajectory $(\mathcal{I}_T, \mathcal{I}_{T-1}, ..., \mathcal{I}_0)$ can be formulated as a state-action trajectory $\tau$ in the MDP, followed by $p(\tau \mid \pi_\theta) = p_\theta(\mathcal{I}_{0:T})$. Now carefully define a reward function $r(\cdot)$ on instances or state-action pairs. Following Eqn. (3), the objective function $\mathcal{J}_{\text{RL}}(\pi)$ is equivalent to the following $\mathcal{J}_{\text{RL}}(\theta)$:

$$\mathcal{J}_{\text{RL}}(\theta) = \mathbb{E}_{p_\theta(\mathcal{I}_{0:T})}[r(\mathcal{I}_0)] \propto -\mathcal{J}_{\text{CR}}(\theta). \quad (6)$$

Next, we introduce how to optimize $\mathcal{J}_{\text{CR}}(\theta)$ using our SPG, which efficiently produces novel instances compared to existing approaches including DDPO.

### 4.3. Shortcut Policy Gradient Algorithm

DDPO optimizes $\mathcal{J}_{\text{CR}}(\theta)$ by estimating the policy gradient $\nabla_\theta \mathcal{J}_{\text{RL}}(\theta)$ with REINFORCE as follows:

$$\nabla_\theta \mathcal{J}_{\text{RL}} = \mathbb{E}\left[\sum_{t=0}^{T} \nabla_\theta \log p_\theta(\mathcal{I}_{t-1} \mid \mathcal{I}_t)\, r(\mathcal{I}_0)\right]. \quad (7)$$

However, in our task, it suffers a poor convergence in experiments and thus fails to generate harder instances. See Figure 4 for an empirical observation. A possible reason can be the tremendous number of trajectories in MDP. Inspired by DDIM (Song et al., 2020), a natural idea is to *skip a few steps* of the denoising process to reduce the trajectory space. To achieve this goal, we slightly modify Eqn. (7) by defining a shortcut policy gradient $S(\theta)$:

$$S(\theta) = \mathbb{E}\left[\sum_{t=0}^{T} r(\mathcal{I}_0)\left(\sum_{m=1}^{M} \nabla_\theta \log p_\theta(\mathcal{I}_{t-mk} \mid \mathcal{I}_{t-(m-1)k})\right)\right], \quad (8)$$

where $k$ is a hyper-parameter that determines the skipping step size, and $M = t/k$[1]. Intuitively, in Eqn. (7) it is overly detail-oriented to focus on how $\mathcal{I}_t$ becomes $\mathcal{I}_0$ through a very long trajectory. The novel harder instances may often lie at the neighbor of the pre-trained $p_\theta$ through our experimental observations. It suffices to skip some steps of the reverse process by sampling a subsequence $[\mathcal{I}_t, \mathcal{I}_{t-k}, \ldots, \mathcal{I}_0]$

---

[1] We choose $k$ that divides $t$.

within $[\mathcal{I}_t, \mathcal{I}_{t-1}, \ldots, \mathcal{I}_0]$ with the step size of $k$. This leads to a much better convergence in our task than DDPO.

Further, computing the gradient $\nabla_\theta \log p_\theta(\mathcal{I}_{t_0-k}|\mathcal{I}_{t_0})$ can be highly costly for a large $k$, due to the chain rule of the derivative. It may cause a GPU memory overflow for large instances. To reduce the computational overhead, we instead estimate the gradient in Eqn. (8) by Eqn. (2) in DDIM. Our experiments will demonstrate that even a very large $k$, for example $k = t$, produces high-quality instances, while highly reducing the overhead in the fine-tuning process. Algoithm 1 presents details of our SPG.

---

**Algorithm 1** Shortcut Policy Gradient Algorithm

---

**Require:** DDPM $p_\theta$, diffusion steps $T$, reward $r(\cdot)$, epoch numbers $E$, trajectory samples $N$, step size $k$, overall steps $M$, learning rate $\eta$

**Ensure:** Updated DDPM $p_\theta$

1: **for** $e = 1, \ldots, E$ **do**
2:     **for** $n = 1, \ldots, N$ **do**
3:         // Sample trajectories and get rewards
4:         $\mathcal{I}_{0:T}^{(n)} \sim p_\theta, r_n \leftarrow r(\mathcal{I}_0^{(n)})$
5:         // Calculate the gradient by Eqn. (2)
6:         $\mathcal{G}_n = \sum_{t=0}^{T} \sum_{m=1}^{M} \nabla_\theta \log p_\theta(\mathcal{I}_{t-mk}^{(n)} \mid \mathcal{I}_{t-(m-1)k}^{(n)})$
7:     **end for**
8:     // Estimate the shortcut policy gradient
9:     $S(\theta) \leftarrow \frac{1}{N} \sum_{n=1}^{N} r_n \cdot \mathcal{G}_n$
10:    // Update model parameter
11:    $\theta \leftarrow \theta + \eta \cdot S(\theta)$
12: **end for**

---

## 5. Experiments

In this section, we first show how to apply DiMa to the classic (fractional) OBM and reproduce the known best $1 - 1/e$ upper bound by generating the known hardest upper-triangular graph instance (Section 5.1). Further, we study two important variants of OBM, named *OBM with random arrivals* (Section 5.2) and *OBM with stochastic rewards* (Section 5.3). Both problems are still open-ended, meaning that their best upper bounds remain unknown. We improve their state-of-the-art by obtaining harder instances by DiMa. Our code is provided in the Supplementary Material.

### 5.1. (Fractional) Online Bipartite Matching

**Model.** The input instance $\mathcal{I} = (L \cup R, E)$ of the fractional OBM is identical to the (integral) OBM introduced in 3.1, but the algorithm is allowed to match the online vertex $r_j$ with a fraction of $x_{ij}$ to an $\ell_i$, under constraints:

$$\sum_{\ell_i \in L} x_{ij} \leq 1, \forall r_j \in R; \text{ and } \sum_{r_j \in R} x_{ij} \leq 1, \forall \ell_i \in L.$$

**Existing theoretical results.** *Water-filling* (Kalyanasundaram & Pruhs, 2000), a simple deterministic greedy algorithm, is known to be optimal for the fractional OBM, achieving $1 - 1/e$-competitive. See Appendix B.1 for the detail of Water-filling. We fine-tune our model using SPG against the Water-filling. The hardness result of Water-filling, and also of the fractional OBM problem, is a classic upper-triangular graph instance as shown in Figure 2(a).

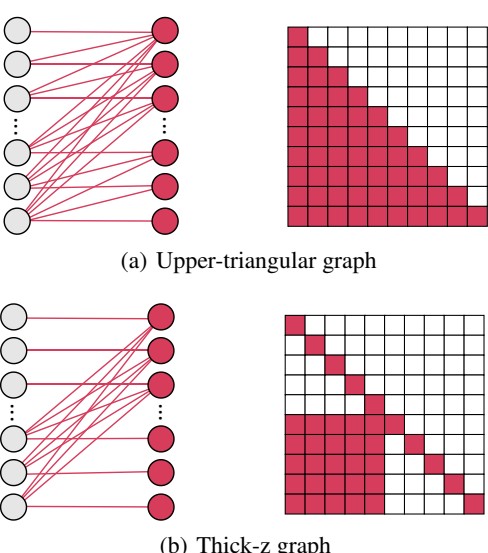

(a) Upper-triangular graph

(b) Thick-z graph

*Figure 2.* (a)The known hardest instance of Water-filling for fractional OBM. (b)A known hard instance of OBM problems, which serves to construct $q$.

**Experiment setup.** We set up DiMa on the fractional OBM as follows:

- **Choice of $q$ in DDPM.** In this problem, $q$ is initialized by a *thick-z* distribution constructed as follows. We construct a set of the thick-z distribution that comprises 200 graphs by adding randomness to the thick-z graph instance as presented in Figure 2, which is known to be difficult (but not the worst) for OBM problems. Instances are obtained by independently flipping each edge in thick-z with a probability of $\gamma = 0.25$. $\gamma$ serves as a hyper-parameter to enhance the diversity of the $q$ distribution. We highlight that determining an appropriate $\gamma$ is crucial in our task. Neither a large nor tiny $\gamma$ produces novel harder instances. See Appendix B.2 for detailed discussion.

- **DDPM structure.** The denoising model is structured with diffusion transformer (DiT) (Peebles & Xie, 2023). DiT is based on the vision transformer (ViT), retaining most of its configurations. In our experiment, we adopt a DiT model with 3 DiT blocks, setting the patch size as 4 and the embedding size as 32.

- **Output format.** Recall that we allow instances $\mathcal{I}$ to be real-valued during DDPM pre-training due to the differentiability. We adopt a standard rounding policy to obtain a 0-1-valued adjacent matrix $\hat{\mathcal{I}}$ as the initial state of fine-tuning:

$$\hat{\mathcal{I}}_{ij} = \begin{cases} 1, & \text{if } \mathcal{I}_{ij} \geq 0.5 \\ 0, & \text{if } \mathcal{I}_{ij} < 0.5 \end{cases} \qquad (9)$$

- **Reward function.** In the fine-tuning process using SPG, the reward function $r(\mathcal{I})$ is defined as the $\lambda(1 - \text{CR}(\mathcal{I}))$, where $\text{CR}(\mathcal{I}) = \frac{\text{ALG}(\mathcal{I})}{\text{OPT}(\mathcal{I})}$ and $\lambda = 5$ is a hyper-parameter that controls the magnitude of the reward. ALG can be computed by running Water-filling on each instance $\mathcal{I}$. OPT can be computed by solving a linear program on $\mathcal{I}$ directly.

- **Other hyper-parameters.** The size of the problem instance $\mathcal{I}$ is set to $|L| = |R| = 20, 40$. In the pre-training process, the total number of time steps is set to $T = 100$, the training batch size is set to 4, and the training epoch is set to 1000. In the fine-tuning process, the total number of training epochs is set to $E = 100$ with $N = 100$ trajectories sampled per epoch and the step size $k$ is set to $k = 100$.

**Experimental results.** We show the effectiveness of our DiMa on the fractional OBM through the following experiments. More experiments are presented in Appendix B.2.

*Experiment #1: DDPM learns the thick-z distribution after pre-training.* The upper half of Figure 3 presents how the instance is sampled in the reverse denoising process through the pre-trained model.

As the time step $t$ changes from 100 to 0, the instance gets progressively closer to the thick-z graph depicted in Figure 2(b). We remark that the pre-trained instance distribution should be similar (but not identical) to the known hardness of the thick-z graph. Otherwise, it fails to explore novel harder instances. Recall that we control such similarity when constructing $q$ and using a randomness hyper-parameter $\gamma$. Figure 11 in Appendix B.2 presents an ablation on $\gamma$.

*Experiment #2: DiMa converges at the hardest upper-triangular distribution after fine-tuning by SPG.* During the fine-tuning process, instances sampled from the pre-trained model are first rounded to an adjacent matrix $\{0, 1\}^{|L| \times |R|}$ according to Eqn.(9). The lower half of Figure 3 presents the learned instances of different epochs. DiMa gradually produces instances approaching the upper-triangular distribution shown in Figure 2(a), and finally converges at the upper-triangular graph after 100 epochs.

*Experiment #3: the effectiveness of SGP.* We compare our SGP to DDPO (Black et al., 2023) in Figure 4. We experiment on two instance sizes $N$ of 20 and 40. While our SPG

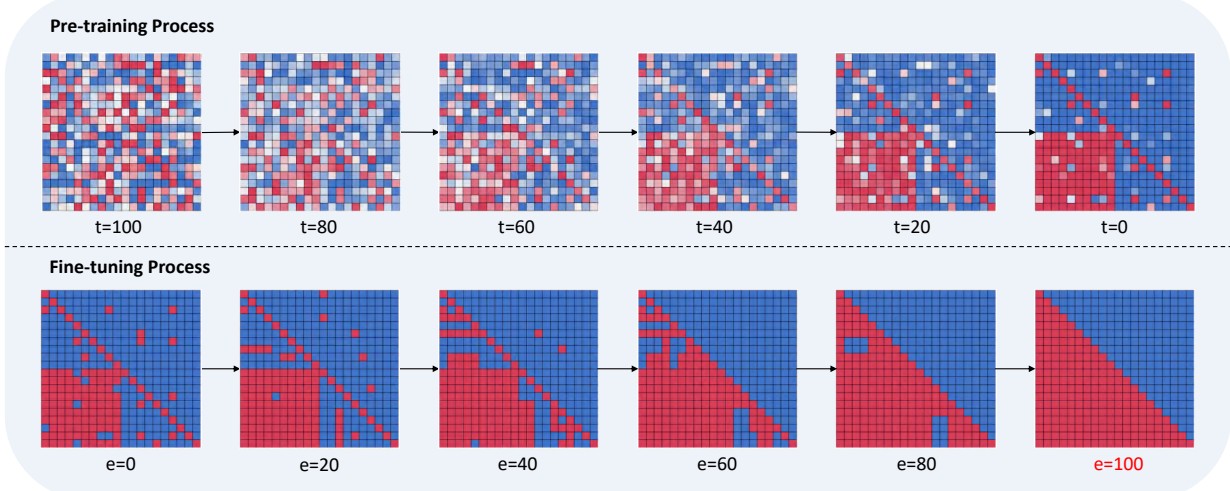

Figure 3. Instances generated from a single trajectory at $t = 100, 80, 60, 40, 20, 0$ during the pre-training process, and hard instances generated by DiMa at different epochs during the fine-tuning process.

obtains an increasing average reward, DDPO suffers a poor convergence and fails to produce any novel instances. Further evaluations on the effectiveness of SPG are presented in Appendix B.2 Figure 9 and Figure 10. We remark on our SGP that we believe SGP is powerful in various downstreaming tasks other than the instance generation in this paper. Nevertheless, further exploring the power of SGP may be of independent interest to the scope of this paper, and thus we leave it as an interesting future direction.

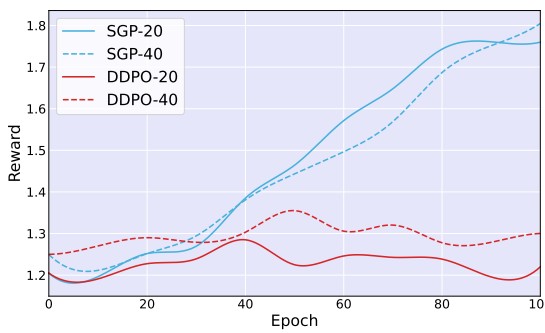

Figure 4. The average reward change of SGP and DDPO during the fine-tuning process with instance sizes of 20 and 40 when sampling.

## 5.2. Online Matching with Random Arrivals

**Model.** In the classic OBM, the algorithm assumes to know nothing about future online vertices in $R$. In online matching with random arrivals (OMRA) (Goel & Mehta, 2008), the algorithm assumes to know the complete graph $\mathcal{I}$, but the arrival order of $R$. $R$ assumes to arrive in a random order. Since the input instances are randomized, the objective is to maximize the *expected* matching size. Note

that the OMRA is easier than OBM because algorithms can get a complete input instance at the beginning.

**Existing theoretical results.** A classic *Ranking* algorithm (Karp et al., 1990) is known to be efficient (but not optimal). Intuitively, Ranking assigns a random permutation to all offline vertices in $L$, and greedily picks one of the unmatched neighbor ranks the highest in the permutation on the arrival of each online vertex. See Appendix C.1 for details of Ranking. The best-known upper bound of Ranking is 0.727 (Karande et al., 2011), which is derived from a hard instance as in Figure 5(a).

**Our results[2].** Figure 5(b) showcases an instance sampled after the pre-training process. We improve the SOTA upper bound of 0.727 to 0.723 by generating new and harder instances (Figure 6) than those given by Karande et al. (2011).

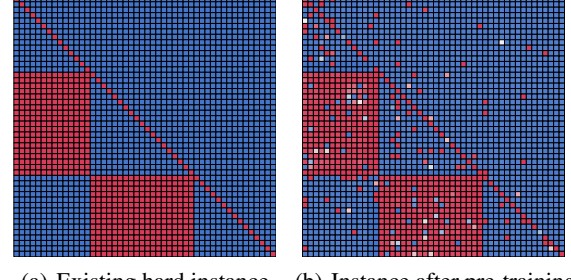

(a) Existing hard instance.  (b) Instance after pre-training.

Figure 5. (a) The known existing hard instance of Ranking for the OMRA problem. (b) Instance sampled by DiMa after pre-training.

**Theorem 5.1.** *There exists a family of instances on which*

---

[2]Experiment setups and detailed experimental results are included in Appendix C.

*Ranking does no better than* $0.723$ *in OMRA.*

*Proof sketch.* The proof (in Appendix F.1) is completed by a careful calculation of the theoretical CR of Ranking running on the new instance in Figure 6.

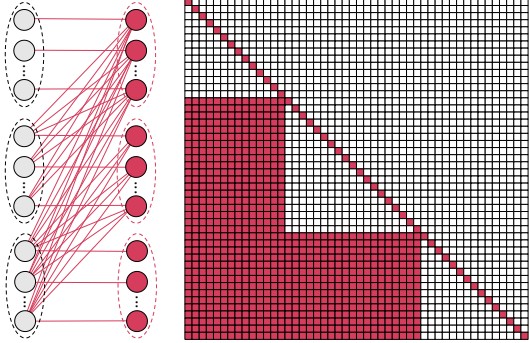

*Figure 6.* Hard instance produced by DiMa that corresponds to a $0.723$ upper bound.

### 5.3. Online Matching with Stochastic Rewards

Online matching with stochastic rewards (Mehta & Panigrahi, 2012) (OMSR) is a strict generalization of classic OBM. Given the instance $\mathcal{I}$, there is a real-number success probability $p_{ij} \in [0, 1]$ on each edge $(\ell_i, r_j) \in \mathbb{E}$. When the algorithm decides to match $r_j$ to $\ell_i$, this match is determined to be *successful with the probability of* $p_{ij}$. If so, $\ell_i$ can not be matched to later coming online vertices. Otherwise, $\ell_i$ is still available. Since the matching process involves randomness, the objective is to maximize the *expected* number of successful matchings. Note that the OBM is a special case of the OMSR by setting all $p_{ij}$ as $\{0, 1\}$.

**Existing theoretical results.** The *Balance* algorithm (Mehta & Panigrahi, 2012) is known to achieve a SOTA CR (but not optimal) for OMSR. Intuitively, Balance matches each online vertex to an unsuccessful neighbor with the least number of failure attempts. See Appendix D.1 for details of Balance. Mehta & Panigrahi (2012) first construct hard instances that derive a $0.621$ upper bound, which is recently improved to $0.597$ by an adversarial RL approach (Zhang et al., 2024a). Their instance is presented in Figure 7(a).

**Our results[3].** Figure 7(b) showcases an instance sampled after the pre-training process. We improve the SOTA upper bound of $0.597$ to $0.594$ by generating new and harder instances (Figure 8) than those given by Zhang et al. (2024a).

**Theorem 5.2.** *There exists a family of instances on which there is no algorithm that does better than* $0.594$ *in OMSR.*

*Proof sketch.* The proof (in Appendix F.2) comprises two key ingredients: to prove that Balance is optimal to OMSR

---

[3]Experiment setups and detailed experimental results are included in Appendix D.

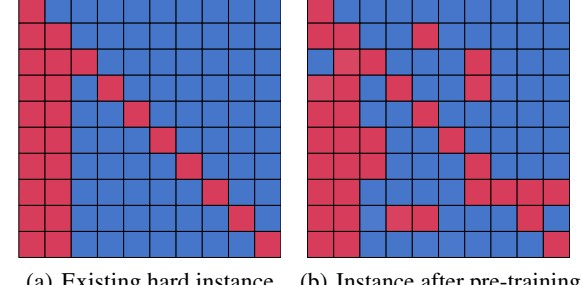

(a) Existing hard instance.    (b) Instance after pre-training.

*Figure 7.* (a) The known existing hard instance of the OMSR problem. (b) Instance sampled by DiMa after pre-training.

on the given instances in Figure 8, and to carefully compute the theoretical CR of Balance running on the instances.

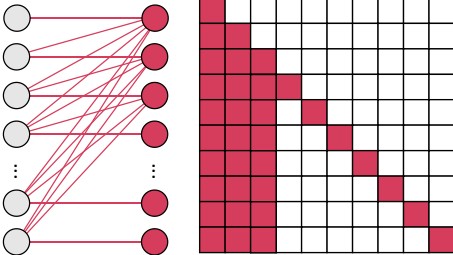

*Figure 8.* Hard instance produced by DiMa that corresponds to a $0.594$ upper bound.

**Remark.** The smallest upper bound in Zhang et al. (2024a) is achieved when $N = 7$. They report an upper bound of $0.599$ when $N = 10$, while we present a better $0.594$ upper bound at $N = 10$. Thus the actual numerical improvement in the upper bound is from $0.599$ to $0.594$, indicating a further effectiveness of our DiMa. Besides, our DiMa can proceed with instances of much larger size than Zhang et al. (2024a), e.g., $N = 48$ in the OMRA problem. However, for ease of proof, we only present an $N = 10$ instance here.

## 6. Conclusion

This paper introduces DiMa, to enhance the theoretical understanding of OBM problems assisted by ML techniques. To the best of our knowledge, this is the first unified framework that generates hard instances used for proving an improved theoretical upper bound of the competitive ratio. We apply DiMa to three fundamental OBM problems in literature. Extensive experiments show that DiMa can not only reproduce the known hardest instances, but also generate novel instances that induce better bounds for those open-ended OBM problems. As an independent technical contribution, we propose a shortcut gradient policy algorithm for optimizing the DDPMs using RL. SPG greatly reduces

the overhead of the gradient calculation, while maintaining the quality of the generated instance. We believe DiMa has great potential in being applied to a broader range of online optimization problems, or even offline combinatorial optimizations, such as approximation algorithms, which we leave as a future direction. Another interesting future work is to explore the ability of other AI artifacts, such as large language models, in optimization theory. We hope that DiMa may inspire more interesting attempts in the area of AI for theoretical computer science in the future.

## Acknowledgements

Qiankun Zhang is supported by the National Natural Science Foundation of China (Grant 62302183), Open Foundation of Key Laboratory of Cyberspace Security, Ministry of Education of China (Grant KLCS20240401), Ant Group Research Fund (Grant 20242452) and CCF-DiDi GAIA Collaborative Research Funds (Grant CCF-DiDi GAIA 202412). Jing Wang is supported by the National Natural Science Foundation of China (Grant 62202197), and the Major Research Project of Hubei Province (Grant 2023BAA027). Bin Yuan is supported by the National Natural Science Foundation of China (Grant 62372191), the Open Topics from The Lion Rock Labs of Cyberspace Security (Grant LRL24013), and Songshan Laboratory (Grant 241110210200). Xianjun Deng is supported by the National Key R&D Program of China (Grant 2022YFE0138600), and the National Natural Science Foundation of China (Grant U24B20153).

## Impact Statement

This paper advances the field of ML by introducing a novel framework, DiMa, that trains a diffusion model to enhance the theoretical understanding of the famous online matching problems. DiMa demonstrates the great potential of ML in theoretical computer science and may inspire future work in other fields of TCS, such as approximation algorithms or algorithmic game theory.

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

## A. Other Related Works

**OBM theory.** OBM problem receives remarkable attention in theoretical computer science since Karp et al. (1990). We refer readers who are interested in but not familiar with theoretical studies of OBM to a survey (Mehta et al., 2013), which records the most significant open problems about OBM, including the upper bounds we study. There are extensive theory works about OBM, but we only discuss those most relevant to our work. For the classic OBM, Karp et al. (1990) provides an optimal $1 - 1/e$-competitive algorithm named Ranking, as well as the hardest instance of the upper-triangular graph. Karande et al. (2011) breaks the $1 - 1/e$ barrier by introducing the random arrival model, and gives a hardness result of $0.727$ against Ranking. To the best of our knowledge, there has been no follow-up work improving this bound since Karande et al. (2011). Other works related to OMRA include (Jin & Williamson, 2021; Fahrbach et al., 2022; Chen et al., 2024; Mahdian & Yan, 2011). For the stochastic rewards problem, Mehta et al. (2013) first introduces it with a $0.621$ upper bound. There is no other work improving this bound until a recent work by Zhang et al. (2024a), who use an RL-assisted approach to find a family of harder instances. Our DiMa not only outperforms them in obtaining harder instances and thus improving the upper bound, but also in efficiency, instance size, and generalizability to a set of OBM problems. Other works related to OMSR include (Huang & Zhang, 2024; Goyal & Udwani, 2023; Huang et al., 2023; Udwani, 2024).

**DDPMs.** DDPMs have emerged as a powerful class of generative models in recent years. The foundational work on DDPMs was pioneered by Sohl-Dickstein et al. (2015), and later simplified by Ho et al. (2020). DDPMs generate samples by adding noise in a forward pass and removing it via a trained network in reverse. Since its inception, the diffusion model has made significant inroads in a variety of downstream tasks, driving remarkable advancements in the field. For example, Ho et al. (2022b) designed hierarchical architectures that are instrumental in stabilizing the training process of diffusion models for image generation and mitigating memory cost issues. Ramesh et al. (2022) integrated the diffusion model into text-to-image generation tasks, with their DALL-E2 model demonstrating outstanding performance in this domain. Apart from image generation (Saharia et al., 2022), they have also made contributions in the fields of video generation (Ho et al., 2022a; Singer et al., 2022), robot guidance (Janner et al., 2022; Jackson et al., 2024) and image-to-3D conversion (Zhou et al., 2021; Xu et al., 2023). Moreover, DDPM has also been widely applied in the medical field (Guha & Acton, 2023; Asgariandehkordi et al., 2023; Xiang et al., 2023).

Recent studies (Sun & Yang, 2023; Sanokowski et al., 2024) have explored the applications of the diffusion model to combinatorial optimization problems by treating the solution space as a probability distribution and leveraging the iterative denoising process to generate high-quality solutions. These works primarily focus on empirical demonstrations, showcasing the effectiveness of diffusion models in specific problem domains. However, their approaches lack theoretical backgrounds and guarantees, making it difficult to directly benchmark against our method, which provides both theoretical proof and comprehensive evaluations.

## B. Fractional Online Bipartite Matching

### B.1. Water-filling Algorithm

**Water-filling algorithm.** When an online vertex $r_j$ arrives, Water-filling allocates an infinitesimal $\mathrm{d}x$ portion of $r_j$ to a neighbor with the largest remaining capacity, until no neighbor is available or $r_j$ is exhausted. Recall that $x_{ij}$ denotes the portion that matches $\ell_i$ to $r_j$ in the fractional matching, and let $y_i = \sum_{j:(\ell_i, r_j) \in E} x_{i,j}$ be the allocated capacity of $\ell_i$. Algorithm 2 is a formal definition of Water-filling.

---

**Algorithm 2** Water-filling

---

1: Initialize all $x_{ij}$s to be zero;
2: **for** each online vertex $r_j$ that arrives **do**
3:     **while** $\sum_{i:(i,j) \in E} x_{ij} < 1$ and $y_i < 1$ for some $i$ s.t. $(i, j) \in E$ **do**
4:         Allocate a $\mathrm{d}x$ portion of $r_j$ to an neighbor in $\arg\max_{i:(i,j) \in E}\{1 - y_i\}$, breaking ties arbitrarily, i.e., increase $x_{ij}$
        by $\mathrm{d}x$.
5:     **end while**
6: **end for**

---

## B.2. Ablation Study

**The impact of skipping step size $k$ in SPG.** We compare the effects for different step sizes $k$ when using SGP during the fine-tuning process. In the fractional OBM problem, we use SGP with different step sizes $k$ during the fine-tuning process.

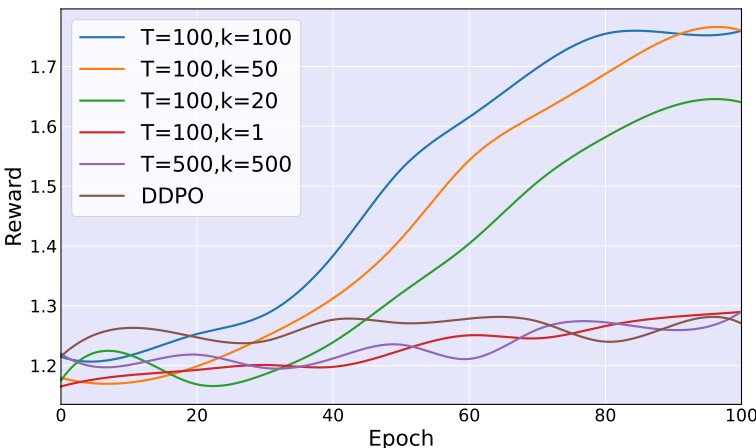

*Figure 9.* The average rewards of sampling instances using SGP during the fine-tuning process when $T = 100$, $k = 1, 20, 50, 100$ and $T = 500$, $k = 500$ and the average rewards of sampling instances using DDPO when $T = 100$.

As shown in Figure 9, we can find that:

- When $k = 1, M = 100$, the average reward of the model during the fine-tuning process shows an extremely slow growth trend and has certain fluctuations which is similar to DDPO.

- The average reward of the model shows a distinct upward trend during the fine-tuning process with the increasing of the step size $k$. Moreover, the rate of increase in the average reward accelerates as the value of $k$ increases. When $k = 100$ and $M = 1$, the model can converge to the upper-triangular distribution within 100 epochs.

- Furthermore, as $k$ increases, the overhead of calculating gradients decreases accordingly. Therefore, in our experimental setup, we choose $k = T = 100$. However, as $T$ increases, when $T = 500$, if we choose $k = 500$, The effectiveness of SPG gets worse due to the poor quality of generation. Thus, it is necessary to select an appropriate $k$ for different values of $T$ to achieve a performance-efficiency trade-off.

**The impact of number of sample trajectories $N$.** We compare the impacts for the number $N$ of sampling trajectories when using SGP during the fine-tuning process. In the fine-tuning process, we adopt SGP and fine-tune it with different numbers of sampling trajectories at each epoch.

As shown in Figure 10, when the hyper-parameter $N$ of SGP is set to 100, the average reward steadily rises during the fine-tuning process and gradually converges. Additionally, the fine-tuning process gradually becomes more unstable with the decrease of $N$. Therefore, we typically set $N$ to 100 to obtain stable training results.

**The impact of $q$.** We compare the influence of the initial data distribution $q$ on SGP. During the pre-training process, we select different hyper-parameter $\gamma$ for the initial data distribution and perform pre-training and fine-tuning respectively.

As shown in Figure 11, we could find that:

- When $r$ is relatively large ($\gamma > 0.50$), the effect of fine-tuning gets worse because of the excessive diversity of the initial data distribution. The initial data distribution deviates excessively from the distribution of the already-known hard instances.

- When $r$ is relatively small ($\gamma < 0.10$), the initial data distribution is more similar to thick-z which leads to higher effects at the beginning of the fine-tuning process. However, we can only get a lower average reward during the fine-tuning process since it does not learn more new situations.

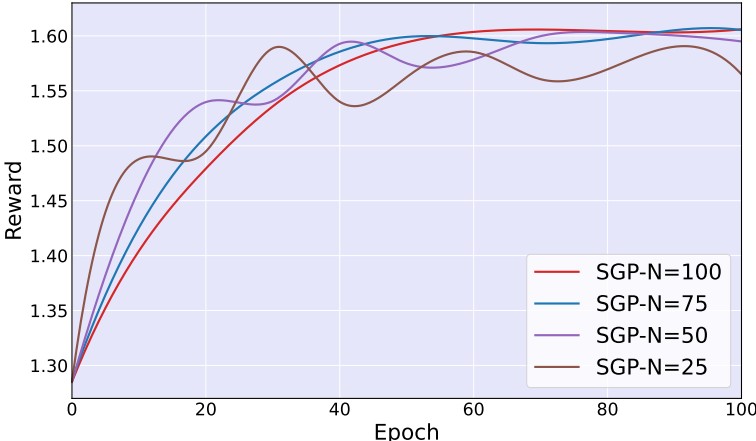

*Figure 10.* The average rewards of sampling instances using SGP during fine-tuning process with different numbers of sampling trajectories $N = 25, 50, 75, 100$.

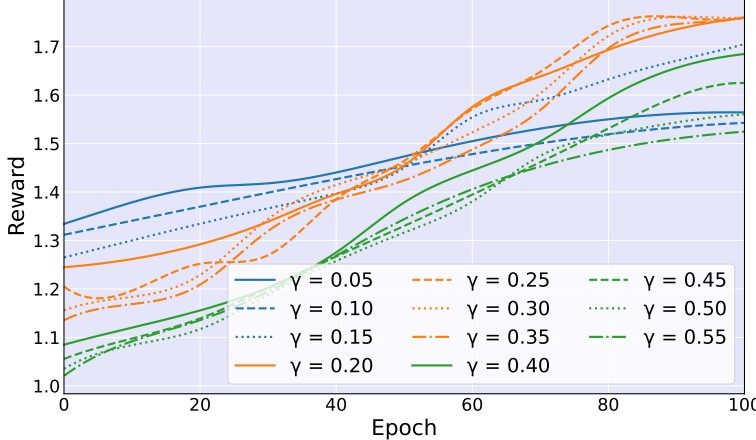

*Figure 11.* The average rewards of sampling instances using SGP during the fine-tuning process with different initial data distributions represented by hyper-parameter $\gamma = 0.05, 0.10, 0.15, 0.20, 0.25, 0.30, 0.35, 0.40, 0.45, 0.50, 0.55$.

- Our DiMa easily converges to the worst upper-triangular instances within 100 epochs when $\gamma \in [0.2, 0.35]$. Therefore, we need to select an appropriate initial data distribution $q$ with hyper-parameter $r$ to find new harder instances.

As demonstrated by previous experimental results, leveraging structural information from known hard instances can accelerate convergence and improve sample efficiency. Such an idea also aligns with conventional hand-crafted constructions in OBM, where new harder instances are typically built on the existing ones with some slight modifications. What's more, we emphasize that DiMa can converge to upper triangulars starting with diverse distributions (not restricted to the thick-z), even with random samples. We visualize some of the initialization distributions that successfully converge to upper triangulars, including randomly sampled ones, which are shown in Figure 12.

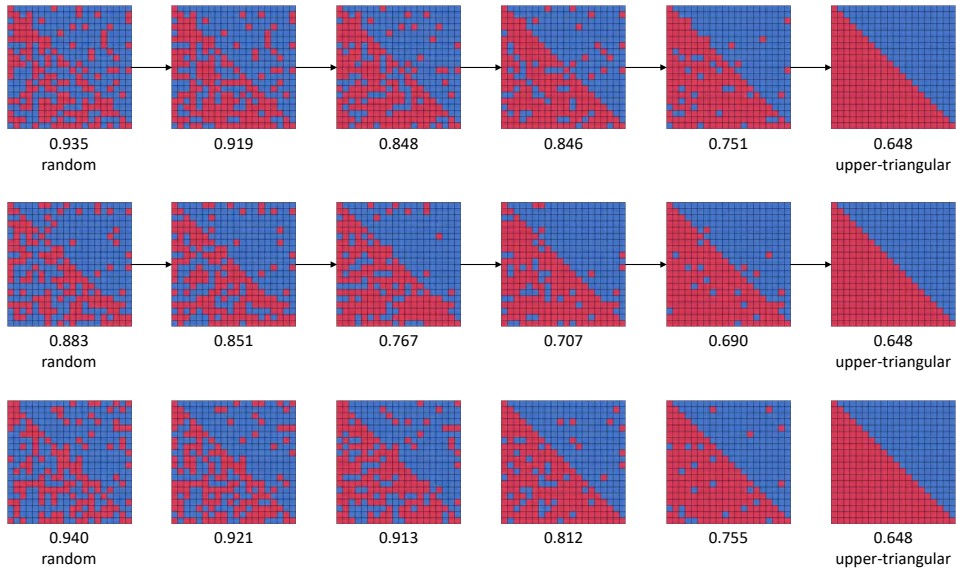

*Figure 12.* The instances with their competitive ratios generated during the fine-tuning process for three randomly initial distributions in the experiments of OBM.

We conduct additional experiments starting with 50 random samples, nearly one-third of which enable DiMa to find the worst instances within 100 epochs. Finally, we emphasize that, unlike Zhang et al. (2024a), which appears to heavily depend on intermediate observations for iterative adjustments during the training process to discover novel instances, our method largely reduces reliance on expert insights because existing hard instances are easily obtained in the literature. Although DiMa can benefit from known hard instances, it is essentially end-to-end.

**The effectiveness of SPG.** We propose SPG as an independent contribution addressing a common challenge in the ML-for-OM literature. Traditional methods struggle with large-scale graphs. For example, in Zhang et al. (2024a), their RL approach for OMSR seems limited to small graphs (fewer than 10 offline vertices). As shown in Table 1, DiMa efficiently works on remarkably larger instances (of size larger than 50) on a 24GB GPU within an hour, and the traditional way to compute the gradient needs more than 24 GB.

*Table 1.* The memory size and running time using SGP during the fine-tuning process when the size of the instance is set to $|L| = |R| = 12, 20, 40, 80$.

| Method | Graph size | Memory size | Running time |
|---|---|---|---|
| SPG | 12x12 | 0.62GB | 20s/epoch |
| SPG | 20x20 | 1.14GB | 25s/epoch |
| SPG | 40x40 | 5.05GB | 35s/epoch |
| SPG | 80x80 | 22.59GB | 45s/epoch |
| Traditional Computation | 20x20 | >24GB | – |

## C. Online Matching with Random Arrivals

### C.1. Ranking Algorithm

**Ranking algorithm.**    Before the first online vertex arrives, Ranking randomly permutes $1, 2, \cdots, |L|$, which determines the ranks of the offline vertices. When an online vertex $r_j$ arrives, Ranking matches $r_j$ to an available neighbor with the highest rank.

---

**Algorithm 3** Ranking

---

 1: Randomly permute the sequence $(1, 2, \cdots, |L|)$ as $\rho$, and let $\rho_i$ be $\ell_i$'s rank;
 2: Initialize all $x_{ij}$s to be zero;
 3: **for** each online vertex $r_j$ that arrives **do**
 4:     **if** there exists an available neighbor of $r_j$ **then**
 5:         Match $r_j$ to $\arg\min_{i:(i,j)\in E}\{\text{rank}_i\}$, i.e., set $x_{ij} = 1$.
 6:     **end if**
 7: **end for**

---

### C.2. Experiment Setup

Here we present the experimental settings in completing the relevant experiments of the OMRA problem. We set up DiMa as follows:

- **Choice of $q$ in DDPM.** In this problem, $q$ is initialized by an existing distribution of the hard instance as follows. We construct a set of the existing distribution that comprises 200 graphs by adding randomness to the hard instance as presented in Figure 5(a), which is known to be difficult for OMRA problems. Instances are abtained by independently flipping each edge in the hard instance with a probability of $r = 0.1$. $r$ serves as a hyper-parameter to enhance the diversity of the $q$ distribution.

- **DDPM structure.** The denoising model is structured with DiT. In our experiment, we adopt a DiT model with 3 DiT blocks, setting the patch size as 4 and the embedding size as 32.

- **Output format.** Recall that we allow instances $\mathcal{I}$'s to be real-valued during DDPM pre-training due to the differentiability. We adopt a standard rounding policy by using Equation (9) to obtain a 0-1-valued adjacent matrix $\hat{\mathcal{I}}$ as the initial state of fine-tuning.

- **Reward function.** In the fine-tuning process using SPG, the reward function $r(\mathcal{I})$ is defined as the $\lambda(1 - \text{CR}(\mathcal{I}))$, where $\text{CR}(\mathcal{I}) = \frac{\text{ALG}(\mathcal{I})}{\text{OPT}(\mathcal{I})}$ and $\lambda = 5$ is a hyper-parameter that controls the magnitude of the reward. the details of computing the CR are in Appendix E.

- **Hyper-parameters.** The size of the problem instance $\mathcal{I}$ is set to $|L| = |R| = 48$. In the pre-training process, the total number of time steps is set to $T = 100$, the training batch size is set to 4, and the training epoch is set to 1000. In the fine-tuning process, the total number of training epochs is set to $E = 100$ with $N = 90$ sampled per epoch and the step size $k$ is set to $k = 100$.

### C.3. Experimental Result

The upper half of Figure 13 presents how the instance is sampled in the reverse denoising process through the pre-trained model. As the time step t changes from 100 to 0, the instance gets progressively closer to the hard instance depicted in Figure 5(a). During the fine-tuning process, instances sampled from the pre-trained model are first rounded to an adjacent matrix $\{0, 1\}^{|L| \times |R|}$ according to Eqn. (9). The lower half of Figure 13 presents the learned instances of different epochs. DiMa gradually produces instances approaching the distribution shown in Figure 6, and finally converges at the graph after 100 epochs.

## D. Online Matching with Stochastic Rewards

In this paper, we focus on a special case of OMSR under the assumption of equal probability, where if there is an edge connecting $\ell_i$ and $r_j$, $p_{ij} = p$, and $p$ is a constant. We also restrict the instance with $|L|$ offline vertices and $|L|$ online vertex

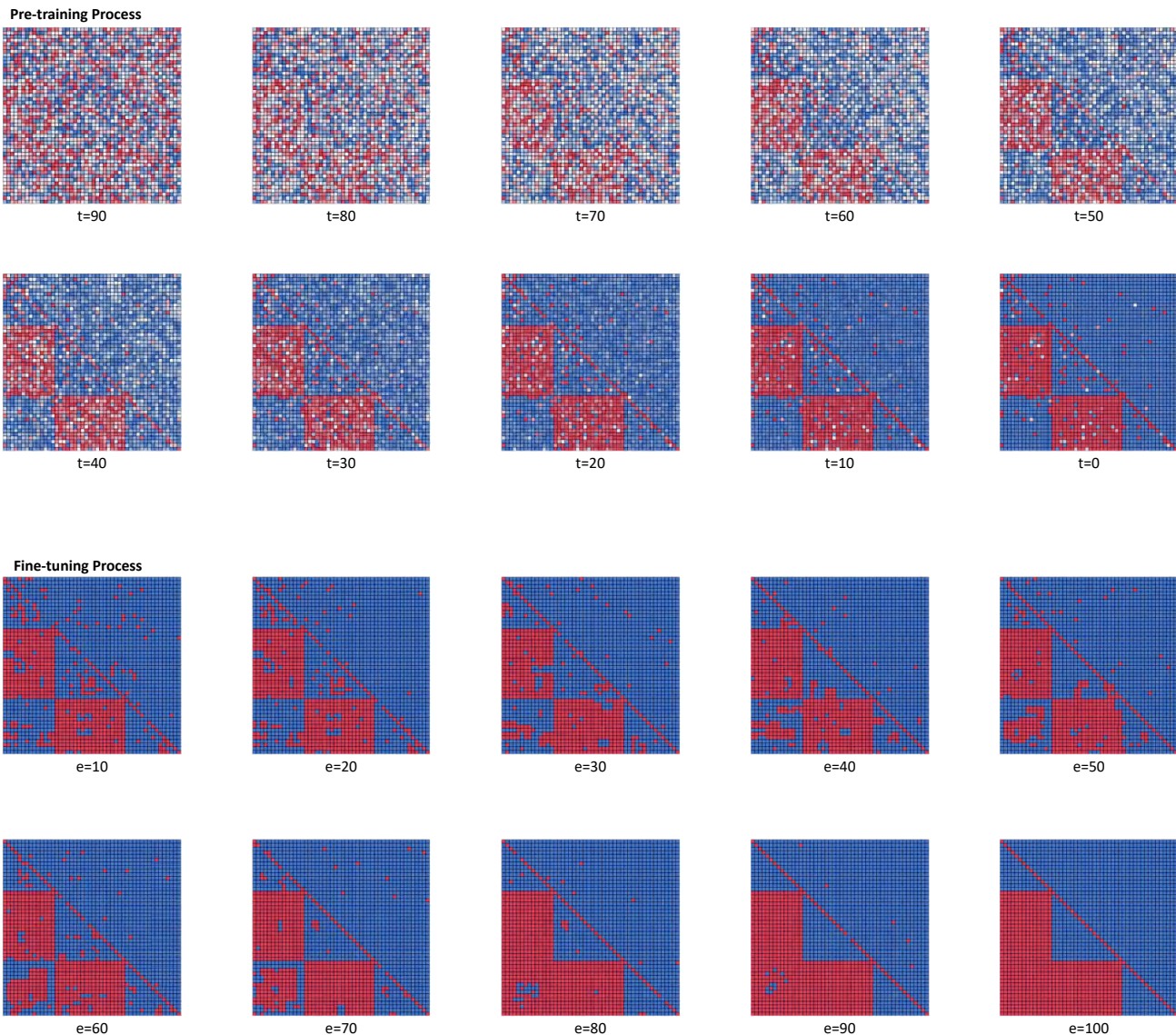

*Figure 13.* Instances of one trajectory generated at t = 90, 80, 70, 60, 50, 40, 30, 20, 10, 0 during the pre-training process and hard instances generated at e = 90, 80, 70, 60, 50, 40, 30, 20, 10, 0 during the fine-tuning process in the experiments of OMRA.

sets to simplify the ratio computation. Each online vertex set has $1/p$ identical offline vertices, sharing the same neighbors. Therefore, $|R| = |L|/p$.

We compare the algorithm against the offline non-stochastic optimum (Mehta & Panigrahi, 2012; Huang et al., 2023; Zhang et al., 2024a). This optimum is the optimal value of a budget allocation problem in the same instance $\mathcal{I}$. In this problem, each edge is associated with a deterministic weight $p$. When the algorithm decides to match $r_j$ with $\ell_i$, $\ell_i$ makes a profit $p$. However, the profit of each offline vertex is capped by 1. The goal of this problem is to maximize the total profit among all offline vertices.

### D.1. Balance Algorithm

**Balance algorithm.** When an online vertex $r_j$ arrives, Balance matches $r_j$ to an available neighbor with the fewest previous failure attempts.

---

**Algorithm 4** Balance

1: Let $\mathrm{cnt}_i$ denote the counter that how many attempts that Balance has made to match $\ell_i$;
2: Initialize all $x_{ij}$s and $\mathrm{cnt}_i$s to be zero;
3: **for** each online vertex $r_j$ that arrives **do**
4:     **if** there exists an available neighbor of $r_j$ **then**
5:         Match $r_j$ a neighbor in $\arg\min_{i:(i,j)\in E}\{\mathrm{cnt}_i\}$, breaking ties arbitrarily, i.e., set $x_{ij} = 1$.
6:     **end if**
7: **end for**

---

### D.2. Experiment setup.

Here we present the experimental settings in completing the relevant experiments of the OMSR problem. We set up DiMa as follows:

- **Choice of $q$ in DDPM.** In this problem, $q$ is initialized by an existing distribution of the hard instance as follows. We construct a set of the existing distribution that comprises 200 graphs by adding randomness to the hard instance as presented in Figure 7(a), which is known to be difficult for OMSR problems. Instances are obtained by independently flipping each edge in the hard instance with a probability of $r = 0.1$. $r$ serves as a hyper-parameter to enhance the diversity of the $q$ distribution.

- **DDPM structure.** The denoising model is structured with DiT. In our experiment, we adopt a DiT model with 1 DiT block, setting the patch size as 2 and the embedding size as 32.

- **Output format.** Recall that we allow instances $\mathcal{I}$'s to be real-valued during DDPM pre-training due to the differentiability. We adopt a standard rounding policy by using Equation (9) to obtain a 0-1-valued adjacent matrix $\hat{\mathcal{I}}$ as the initial state of fine-tuning.

- **Reward function.** In the fine-tuning process using SPG, the reward function $r(\mathcal{I})$ is defined as the $\lambda(1 - \mathrm{CR}(\mathcal{I}))$, where $\mathrm{CR}(\mathcal{I}) = \frac{\mathrm{ALG}(\mathcal{I})}{\mathrm{OPT}(\mathcal{I})}$ and $\lambda = 5$ is a hyper-parameter that controls the magnitude of the reward. the details of computing the CR are in Appendix E.

- **Hyper-parameters.** The size of the problem instance $\mathcal{I}$ is set to $|L| = |R| = 10$. In the pre-training process, the total number of time steps is set to $T = 100$, the training batch size is set to 4, and the training epoch is set to 1000. In the fine-tuning process, the total number of training epochs is set to $E = 100$ with $N = 100$ sampled per epoch and the step size $k$ is set to $k = 100$.

### D.3. Experimental results.

The upper half of Figure 14 presents how the instance is sampled in the reverse denoising process through the pre-trained model. As the time step t changes from 100 to 0, the instance gets progressively closer to the hard instance depicted in Figure 7(a). During the fine-tuning process, instances sampled from the pre-trained model are first rounded to an adjacent matrix $\{0,1\}^{|L|\times|R|}$ according to Eqn. (9). The lower half of Figure 14 presents the learned instances of different epochs.

DiMa gradually produces instances approaching the distribution shown in Figure 8, and finally converges at the graph after 100 epochs.

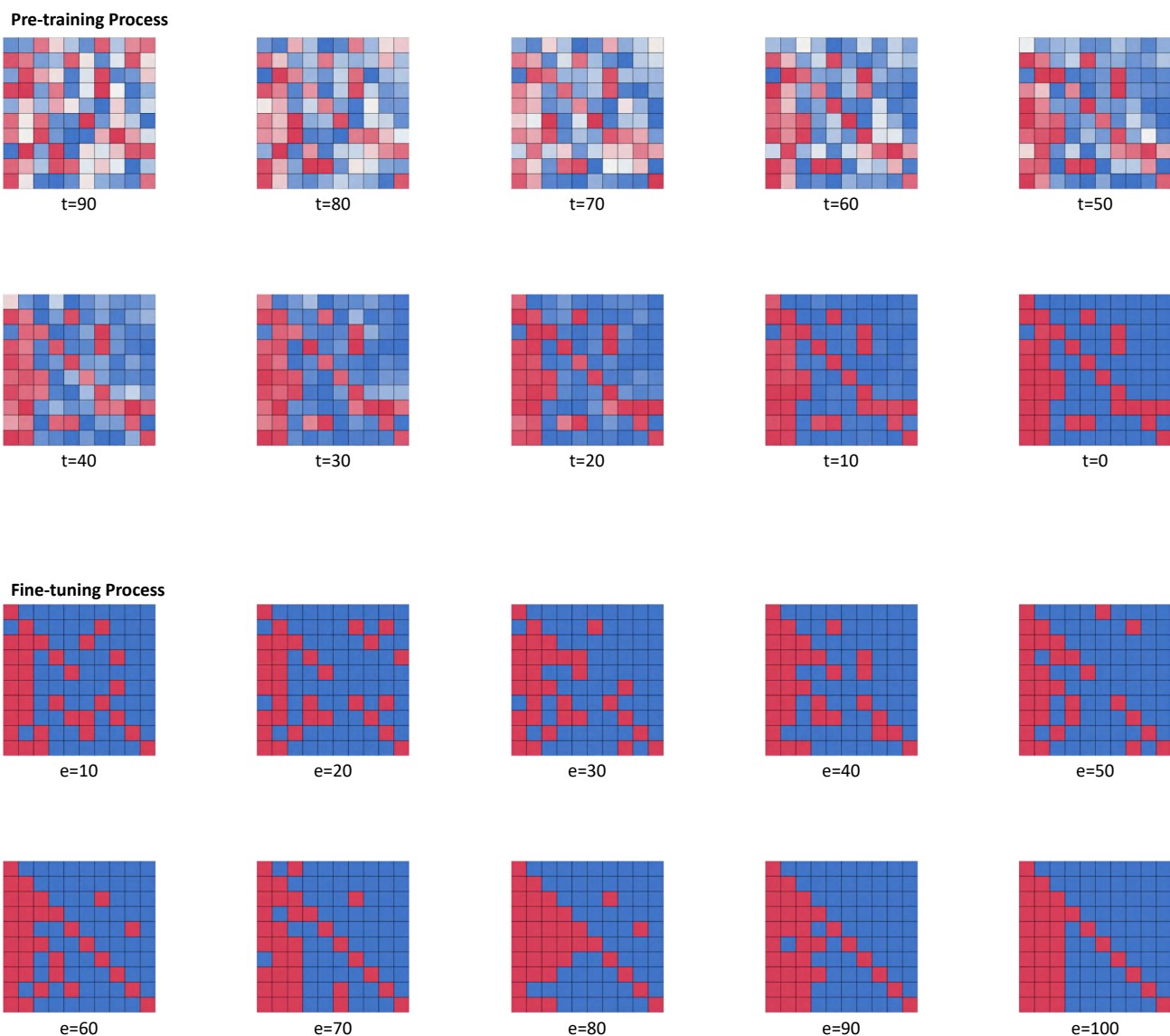

*Figure 14.* Instances of one trajectory generated at t = 90, 80, 70, 60, 50, 40, 30, 20, 10, 0 during the pre-training process and hard instances generated at e = 90, 80, 70, 60, 50, 40, 30, 20, 10, 0 during the fine-tuning process in the experiments of OMSR.

# E. Competitive Ratio Calculation

This section provides a CR calculator to compute the CR of each instance learned by DiMa. Specifically, given an instance $\mathcal{I}$, the calculator returns an estimated value of $\frac{\mathbf{ALG}(\mathcal{I})}{\mathbf{OPT}(\mathcal{I})}$.

### E.1. Calculation on $\mathbf{OPT}(\mathcal{I})$

**Fractional OBM.** The Fractional OBM can be formulated as a linear program to maximize $\sum_{(\ell_i, r_j) \in E} x_{ij}$, subject to the constraints in Eqn. (5.1). Therefore, we use a `simplex` method to solve the LP and return the value $\mathbf{OPT}(\mathcal{I})$.

**OMRA.** In this case, $\mathbf{OPT}(\mathcal{I})$ equals the cardinality of a maximum matching. We use a `dinic` algorithm to compute the value of $\mathbf{OPT}(\mathcal{I})$.

**OMSR.** Recall that we restrict generated instances with $|L|$ offline vertices and $|L|$ online vertex sets. To avoid massive samples of outcome that an edge succeeds or not, we restrict that there are edges connecting $\ell_i$ and $V_i, \forall i \leq |L|$. Therefore, $\mathbf{OPT}(\mathcal{I}) = |L|$.

### E.2. Calculation on $\mathbf{ALG}(\mathcal{I})$

**Fractional OBM.** We discretize the Water-filling process into a step-by-step simulation by defining $\alpha$ as a small fixed value, i.e., $\mathrm{d}x$, then each time we allocate $\alpha$ to a neighbor with the largest remaining capacity. We set $\alpha = 10^{-3}/\min\{|L|, |R|\}$ for experiments.

**OMRA.** We sample $T$ random permutations and simulate the Ranking algorithm on $\mathcal{I}$ with the ranks. The value of $\mathbf{ALG}(\mathcal{I})$ is estimated by averaging the ratios among $T$ samples. We set $T \approx 2 \cdot 10^4 \left(\min\{|L|, |R|\}\right)^2$ for experiments, and this bound is derived by Chebyshev's inequality.

**OMSR.** The procedure executed by the Balance algorithm on $\mathcal{I}$ can be formed as a Markov decision process (MDP) with deterministic actions but stochastic transitions. To compute the value of $\mathbf{ALG}(\mathcal{I})$, we adopt a dynamic programming approach. Specifically, the MDP consists of $|R|$ timesteps. We start with calculating the ratios of different states of offline vertices at the $|R|$-th timestep, using these results to determine the ratios at the $(|R|-1)$-th timestep. This process is iteratively applied until the ratio at the first timestep is obtained and returns the ratio as the value of $\mathbf{ALG}(\mathcal{I})$.

## F. Omitted Proofs

### F.1. Proof of Theorem 5.1

**Characterization on the hard instance.** The instance in Figure 6 has $n$ offline vertices and $n$ online vertices, i.e., $|L| = |R| = n$. The vertices in $L$ are divided into 3 disjoint subsets $L_1$, $L_2$, and $L_3$. Similarly, the vertices in $R$ are also divided into 3 disjoint subsets $R_1$, $R_2$, and $R_3$. The sizes of $L_1, L_3, R_1, R_3$ are $\alpha n, \alpha < 1$, and therefore the sizes of $L_2$ and $R_2$ are $(1-2\alpha)n$. Each offline vertex, $\ell_i$ is connected to the online vertex with the same index $r_i$. In addition, vertices in $R_1$ have edges adjacent to vertices in $L_2$ and $L_3$, and vertices in $R_2$ have edges adjacent to vertices in $L_3$. Note that the online vertices are shuffled according to a uniform random permutation because of the random arrival model. Let $\mathcal{H}$ denote the family of these hard instances in Figure 6.

We first compute the value of $\mathbf{OPT}(\mathcal{H})$. There is a perfect matching in $\mathcal{H}$, which matches $\ell_i$ with $r_i$, for $i \in [n]$. Therefore, $\mathbf{OPT}(\mathcal{H}) = n$. For any $v \in L \cup R$, let $v^*$ denote the adjacent vertex of $v$ in the perfect matching.

Without loss of generality, we assume that one online vertex arrives per union time. At time $t$, an online vertex comes, and let $L_i(t), i = 1, 2, 3$ denote the set of unmatched vertices in $L_i$. Similarly, let $R_i(t), i = 1, 2, 3$ denote the set of unmatched vertices in $R_i$. The vertices in $R_i(t)$ have either yet to arrive or were left unmatched upon arrival. By Lemma F.1, we have $\mathbb{E}[|R_1|] = \mathbb{E}[|L_3|], \mathbb{E}[|R_2|] = \mathbb{E}[|L_2|]$, and $\mathbb{E}[|R_3|] = \mathbb{E}[|L_1|]$ for any $t$.

**Lemma F.1** (c.f. Lemma 2 of Karande et al. (2011)). *For a given graph $G = (L \cup R, E)$ and for a fixed rank vector $\rho$, and the online order vector $\pi$, the output of Ranking with $\rho$ and $\pi$ on $G$ is the same as the output of Ranking with rank vector $\pi$ and order vector $\rho$ on $G' = (R \cup L, E)$.*

Note that any vertex $v$ in $R_1$ can be matched at least by $v^* \in L_1$, but vertices in $R_2$ and $R_3$ are not surely matched. Therefore the ratio of Ranking on $\mathcal{H}$ is

$$\frac{\mathbb{E}\left[\mathbf{ALG}(\mathcal{H})\right]}{\mathbf{OPT}(\mathcal{H})} = 1 - \mathbb{E}\left[\frac{|R_2(n)| + |R_3(n)|}{n}\right] = 1 - \mathbb{E}\left[\frac{|L_2(n)| + |R_3(n)|}{n}\right]. \tag{10}$$

In Lemma F.2, we present the expected decrement of $|R_3(t)|$ in terms of the expected value of $|L_3(t)|$.

**Lemma F.2.** *When $n \to \infty$, for any $t$, we have,*

$$\mathbb{E}[|R_3(t+1)| - |R_3(t)|] = -\frac{\mathbb{E}[|L_3(t)|]}{n - t}. \tag{11}$$

*Proof.* At time $t$, one of the vertices in $R_3(t)$ gets matched (suppose it is $v$) if and only if $v^*$ arrives and $v$ has been matched yet. At this time, the vertex in $R_3$ arrives with probability $\alpha$. In addition, by Chebyshev's inequality, before $t$, $\alpha \cdot t$ vertices in $R_3$ have arrived, and there are $|L_3(t)|$ vertices in $L_3$ have not been matched. Therefore, the probability that $v$ gets matched at $t$ is $\frac{|L_3(t)|}{\alpha(n-t)}$. Therefore, the decrement is $\alpha \cdot \frac{|L_3(t)|}{\alpha(n-t)} = \frac{|L_3(t)|}{n-t}$. $\qquad\square$

Next, we show the expected decrement of $|L_3(t)|$ and $|L_2(t)|$ in Lemma F.3 and F.4. Before the next two lemmas, we introduce some notations here borrowed from Karande et al. (2011). Let $K(t)$ denote the set of vertices, $v$ such that $v \in L_2(t)$ and $v^*$ have not arrived yet. Let $\text{size}L_3(t) = n - \min\{\rho(v)|v \in L_3(t)\}$, $\text{size}L_2(t) = n - \min\{\rho(v)|v \in L_2(t)\}$, and $\text{last}(t) = \max\{\rho(v)|v \in L_2, v \text{ gets matched to a vertex in } R_1\}$. Note that, the rank of $v$ is higher, value of $\rho(v)$ is lower.

**Lemma F.3.** *When $n \to \infty$, for any $t$, we have,*

$$\mathbb{E}[|L_3(t+1)| - |L_3(t)|] = -\frac{\mathbb{E}|L_3(t)|}{n-t} - \left((1-2\alpha)\left(1 - \frac{|K(t)|}{(1-2\alpha)(n-t)}\right) + \frac{|K(t)|}{(1-2\alpha)(n-t)} \cdot \frac{|\text{size}L_3(t)|}{n-|\text{last}(t)|}\right)$$
$$- \alpha \cdot \mathbf{Pr}(\text{size}L_2(t) < \text{size}L_3(t)) \cdot \frac{|\text{size}L_3(t)|}{n}. \tag{12}$$

*Proof.* There are three cases that a vertex in $L_3$ can get matched at time $t$, depending on which set the $t$-th online vertex belongs to. Let the $t$-th online vertex be $v$.

If $v \in R_3$, this happens with probability $\alpha$. In this case, $v$ gets matched if and only if $v^*$ is still available, and this happens with $\frac{L_3(t)}{\alpha(n-t)}$. Therefore, in this case, the probability that vertex in $L_3$ gets matched is $\frac{L_3(t)}{n-t}$.

If $v \in R_2$, this happens with probability $1 - 2\alpha$. In this case, $v$ gets matched if and only if (i) $v^*$ is matched; (ii) $v^*$ is still available and there exists a vertex in $L_3(t)$, whose rank is higher than $v^*$'s. By the definition of $K(t)$, case (i) happens with probability $1 - \frac{K(t)}{(1-2\alpha)(n-t)}$, because there are $(1-2\alpha)(n-t)$ vertices in $R_2$ that have not arrived. Next, compute the probability of case (ii). The probability that $v^*$ is still available is $\frac{K(t)}{(1-2\alpha)(n-t)}$. Note that, $v^*$ is still unmatched, therefore, its rank is lower than $\text{last}(t)$. And when there exists a vertex in $L_3(t)$, whose rank is higher than $v^*$'s, it happens with probability $\frac{\text{size}L_3(t)}{n-\text{last}(t)}$. Thus, when $v \in R_2$, the probability that vertex in $L_3$ gets matched overall is $(1-2\alpha)\left(1 - \frac{K(t)}{(1-2\alpha)(n-t)}\right) + \frac{K(t)}{(1-2\alpha)(n-t)} \cdot \frac{\text{size}L_3(t)}{n-\text{last}(t)}$.

If $v \in R_1$, this happens with probability $\alpha$. In this case, $v$ gets matched if and only if the highest rank in $L_3(t)$ is higher than that in $L_2(t)$ and (i) $v^*$ is matched; (ii) $v^*$ is still available and there exists a vertex in $L_3(t)$, whose rank is higher than $v^*$'s. However, $v^*$ can not be matched before the arrival of $v$. Therefore, case (i) can not happen. The probability that $v^*$ is still available and there exists a vertex in $L_3(t)$, whose rank is higher than $v^*$'s is $\text{size}L_3(t)$. $\qquad\square$

**Lemma F.4.** *When $n \to \infty$, for any $t$, we have,*

$$\mathbb{E}[|L_2(t+1)| - |L_2(t)|] = -\frac{|K(t)|}{n-t} \cdot \frac{|\text{size}L_3(t)|}{n-|\text{last}(t)|} - \alpha \cdot \frac{n-|\text{last}(t)|}{n} \tag{13}$$

*Proof.* There are two cases that a vertex in $L_2$ can get matched at time $t$, depending on the $t$-th online vertex belongs to which set. Let the $t$-th online vertex be $v$.

If $v \in R_2$, this happens with probability $1 - 2\alpha$. In this case, $v$ gets matched if and only if $v^*$ is still available and $v$ can not matched with a vertex in $L_3(t)$. $v^*$ is still available happens with $\frac{K(t)}{(n-2\alpha)(n-t)}$. In addition, the event $v$ can not matched with a vertex in $L_3(t)$ happens if and only if the rank of $v^*$ is higher than all vertices' rank in $L_3(t)$. Therefore, in this case, the probability that vertex in $L_2$ gets matched is $(n-2\alpha) \cdot \frac{K(t)}{(n-2\alpha)(n-t)} \cdot \frac{\text{size}L_3(t)}{n-\text{last}(t)}$.

If $v \in R_3$, this happens with probability $\alpha$. In this case, a vertex in $L_2$ gets matched if and only if the highest rank in $L_2$ is higher than the rank of $v^*$. This happens with probability $\frac{n-\text{last}(t)}{n}$. $\qquad\square$

Lemma F.5 describes the expected behavior of $K$.

**Lemma F.5** (c.f. Lemma 17 of Karande et al. (2011))**.** *When $n \to \infty$, for any $t$, we have,*

$$\mathbb{E}[|K(t+1)| - |K(t)|] = -\frac{|K(t)|}{n-t} - \alpha \cdot \frac{|K(t)|}{|L_2(t)|} \cdot \frac{n-|\text{last}(t)|}{n}. \tag{14}$$

Therefore, by Lemma F.2-F.5 and $\mathbb{E}[|R_1|] = \mathbb{E}[|L_3|]$, $\mathbb{E}[|R_2|] = \mathbb{E}[|L_2|]$, $\mathbb{E}[|R_3|] = \mathbb{E}[|L_1|]$, we apply Kurtz's theorem (Kurtz, 1970), and when $\alpha = 0.28$, we derive $\frac{\mathbb{E}[\mathbf{ALG}(\mathcal{H})]}{\mathbf{OPT}(\mathcal{H})} \approx 0.723$.

## F.2. Proof of Theorem 5.2

**Characterization on the hard instance.** The instance in Figure 8 has 10 offline vertices and 10 online vertex sets, i.e., $|L| = 10$. Each online vertex set comprises $1/p$ vertices, sharing the same neighbors. Let $V_i$ denote the $i$-th online vertex set. The vertices in the first online vertex set $V_1$ are adjacent to all offline vertices. The vertices in $V_2$ are adjacent to all offline vertices except $\ell_1$. The vertices in $V_3$ are adjacent to all offline vertices except $\ell_1$ and $\ell_2$. The vertices in the rest online vertex set have edges only with the corresponding offline vertex, i.e., vertices in $V_j, j \geq 4$ are only adjacent to $\ell_j$. All the probabilities on the edges equal $p$, and $p \to 0$. The order of offline vertex sets is permuted, and let $\mathcal{G}$ denote the family of these hard instances.

To prove Theorem 5.2, it comprises two parts. The first is to show when Balance runs on $\mathcal{G}$, the expected ratio is less than 0.594 (Lemma F.11). The second is to prove that no algorithm is better than Balance on $\mathcal{G}$ (Lemma F.12).

We first compute the expected number of successful matches, i.e., $\mathbb{E}[\mathbf{ALG}(\mathcal{G})]$. By the definition of $\mathbf{ALG}(\mathcal{G})$, the expected value equals the sum of probability that each offline vertex $\ell_i, i \in [|L|]$ succeeds after the execution of Balance on $\mathcal{G}$. Then, we have

$$\mathbb{E}[\mathbf{ALG}(\mathcal{G})] = \sum_{i=1}^{|L|} \mathbf{Pr}(\ell_i \text{ succeeds}). \tag{15}$$

Because the vertices $\ell_i$, where $i \geq 4$, are symmetric, the probabilities of their success are equal. Therefore, it suffices to compute the probabilities of $\ell_i, i \leq 4$.

Next, we show how Balance matches on $\mathcal{G}$. Suppose that when an online vertex set $V_i$ comes, there are $n$ offline vertices available. Lemma F.6 shows the expected number of matches during the arrival of $V_i$.

**Lemma F.6.** *Let the random variable $X_i, 1 \leq i \leq n$ denote the number of successful matches belonging to $V_i$. We have $X_i \sim \min\{\text{Poisson}(1), n\}$, that is,*

$$\mathbf{Pr}(X_i = k) = \begin{cases} \frac{e^{-1}}{k!} & k < n, \\ 1 - \sum_{i=0}^{n-1} \frac{e^{-1}}{i!} & k = n. \end{cases} \tag{16}$$

*Proof.* Balance is an opportunistic algorithm that never leaves an online vertex unmatched if there is an available neighbor. Therefore, when $k < n$, the probability that $k$ matches succeed is that

$$\mathbf{Pr}(X_i = k) = \binom{|V_i|}{k} p^k (1-p)^{|V_i|-k} \tag{17}$$

$$= \frac{|V_i|^{\underline{k}}}{k!} p^k (1-p)^{|V_i|-k} \tag{18}$$

$$= \frac{1}{k!} \left((1/p)^{\underline{k}} \cdot p^k\right)(1-p)^{1/p-k} \qquad (|V_i| = 1/p) \tag{19}$$

$$= \left([1/p]^{\underline{k}} \cdot p^k\right)(1-p)^{-k} \cdot \frac{1}{k!}(1-p)^{1/p} \tag{20}$$

$$= \frac{e^{-1}}{k!}. \qquad (p \to 0) \tag{21}$$

Since there are $n$ vertices available, when $k = n$, it holds

$$\mathbf{Pr}(X_i = n) = 1 - \sum_{i=1}^{n-1} \mathbf{Pr}(X_i = i) \tag{22}$$

$$= 1 - \sum_{i=0}^{n-1} \frac{e^{-1}}{i!}. \tag{23}$$

$\square$

Let $p_k^{(n)}$ denote the value of $\mathbf{Pr}(X_i = k)$, when there are $n$ offline vertices available. Therefore, the probability that $\ell_1$ succeeds can be computed by Lemma F.7.

**Lemma F.7.** *Let the random variable $X_i, 1 \le i \le n$ denote the number of successful matches belonging to $V_1$. The probability that $\ell_1$ succeeds is*

$$\mathbf{Pr}(\ell_1 \text{ succeeds}) = \frac{1}{|L|}\mathbb{E}[X_i] = \frac{1}{|L|}\left(\sum_{k=0}^{|L|} p_k^{(|L|)} \cdot k\right). \tag{24}$$

*Proof.* During the arrivals of the vertices in $V_1$, the probability that each offline vertex succeeds is equal, because they are symmetric. In addition, note that $X_i$ denotes the number of successful offline vertices. Therefore,

$$\mathbb{E}[X_i] = |L| \cdot \mathbf{Pr}(\ell_1 \text{ succeeds}), \tag{25}$$

which finishes the proof. $\qquad\square$

With Eqn. (16), we can compute Eqn. (24) directly. However, the probability that $\ell_i, i \ge 2$ succeeds should be carefully computed by summing the probabilities in all cases of how the online vertices in $V_1, \cdots, V_i$ are matched.

Let $f(i, k)$ denote the probability that there are $k$ offline vertices that succeed, before the arrival of the $(i+1)$-th online vertex set. Therefore, by definition, $f(1, k) = p_k^{(|L|)}$. Lemma F.8 tells us how to recursively compute the value of $f(i, k)$, when $i = 2, 3$.

**Lemma F.8.** *When $i = 2, 3$, let $t = |L| - i + 1$ denote the number of neighbors of $V_i$, and we have*

$$f(i, k) = \sum_{j=0}^{k} f(i-1, j) \cdot \frac{t+1-j}{t+1} \cdot p_{k-j}^{(t-j)} + \sum_{j=0}^{k+1} f(i-1, j) \cdot \frac{j}{t+1} \cdot p_{k-j+1}^{(t-j+1)}. \tag{26}$$

*Proof.* Suppose that there are $j$ vertices that succeed before the arrival of $V_i$. Then, there are two cases on the $j$ vertices when $V_i$ arrives.

In the first case, all the $j$ vertices are the neighbors of $V_i$. This happens with probability $\binom{t}{j}/\binom{t+1}{j} = \frac{t!j!(t+1-j)!}{j!(t-j)!(t+1)!} = \frac{t+1-j}{t+1}$. In this case, $t - j$ vertices remain. If there are $k$ vertices that succeed after the arrival of $V_i$, vertices in $V_i$ should match successfully with $k - j$ offline vertices, and the probability is $p_{k-j}^{(t-j)}$. Hence, the probability overall is

$$f(i-1, j) \cdot \frac{t+1-j}{t+1} \cdot p_{k-j}^{(t-j)}. \tag{27}$$

Sum the probability on different $j$, which forms the first term on the right hand side in Eqn. (26).

If there is one among $k$ vertex which is not $V_i$'s neighbor, this happens with probability $\binom{t}{j-1}/\binom{t+1}{j} = \frac{t!j!(t+1-j)!}{(j-1)!(t-j+1)!(t+1)!} = \frac{j}{t+1}$. In this case, $t - j + 1$ vertices remain. If there are $k$ vertices that succeed after the arrival of $V_i$, vertices in $V_i$ should match successfully with $k - j + 1$ offline vertices, and the probability is $p_{k-j+1}^{(t-j+1)}$. Hence, the probability overall is

$$f(i-1, j) \cdot \frac{j}{t+1} \cdot p_{k-j+1}^{(t-j+1)}. \tag{28}$$

Sum the probability on different $j$, which forms the second term on the right hand side in Eqn. (26). $\qquad\square$

**Lemma F.9.** *The probability that $\ell_i, i = 2, 3$ succeeds is*

$$\mathbf{Pr}(\ell_i \text{ succeeds}) = \frac{1}{t_i}\left(\sum_{k=0}^{t_i} f(i, k) \cdot k\right), \tag{29}$$

*where $t_i = |L| - i + 1$ is the number of $V_i$'s neighbors.*

By the definition of $f$, this proof is similar to the proof of Lemma F.7.

Finally, we show the probability that $\ell_4$ is successful as in Lemma F.10.

**Lemma F.10.** *The probability that $\ell_4$ succeeds is*

$$\mathbf{Pr}(\ell_4 \text{ succeeds}) = \mathbf{Pr}(\ell_3 \text{ succeeds}) + (1 - \mathbf{Pr}(\ell_3 \text{ succeeds}))\left(1 - \frac{1}{e}\right). \tag{30}$$

*Proof.* After the arrival of $V_3$, $\mathcal{G}$ is divided into 7 identical components. Each component is an online vertex set connected to one offline vertex. Consider the reduced graph with $\ell_4$ and $V_4$. Therefore, $\ell_4$ succeeds if and only if it succeeds with the first 3 online vertex sets or it matches successfully with $V_4$. The probability of the first case is $\mathbf{Pr}(\ell_3 \text{ succeeds})$, while the latter is $(1 - \mathbf{Pr}(\ell_3 \text{ succeeds}))(1 - 1/e)$. It holds because vertices in $V_4$ match with $\ell_4$ successfully with probability $p_1^{(1)} = 1 - 1/e$. $\square$

**Lemma F.11.** *When Balance runs on $\mathcal{G}$, the expected ratio of $\mathbf{ALG}(\mathcal{G})$ to $\mathbf{OPT}(\mathcal{G})$ is less than $0.594$, that is,*

$$\frac{\mathbb{E}\left[\mathbf{ALG}(\mathcal{G})\right]}{\mathbf{OPT}(\mathcal{G})} < 0.594. \tag{31}$$

*Proof.* First, we compute the offline optimum. In $\mathcal{G}$, each offline vertex $\ell_i, i \in [|L|]$ is connected with the online vertex set $V_i$. Therefore, $\mathbf{OPT}(\mathcal{G}) = |L|$, as the optimal assignment is to match vertices in $V_i$ to $\ell_i$.

To compute the expected performance of Balance on $\mathcal{G}$, by Eqn. (15), it suffices to sum the probability of each offline vertex up. The expected value of $\mathbf{ALG}(\mathcal{G})$ is $\mathbf{Pr}(\ell_1 \text{ succeeds}) + \mathbf{Pr}(\ell_2 \text{ succeeds}) + \mathbf{Pr}(\ell_3 \text{ succeeds}) + 7 \cdot \mathbf{Pr}(\ell_4 \text{ succeeds})$.

By Lemma F.7, F.9, and F.10, we can compute the probabilities using a dynamic programming method. Therefore, it can derive $\frac{\mathbb{E}[\mathbf{ALG}(\mathcal{G})]}{\mathbf{OPT}(\mathcal{G})} = \frac{(-6103798e^3 - 84712761 + 1413664e^2 + 34811393e + 7257600e^4)e^{-4}}{9072000} \approx 0.59358969 < 0.594.$ $\square$

The last, we need show no algorithm is better than Balance on $\mathcal{G}$. Actually, $\mathcal{G}$ has the consistency property and the exclusivity property (Zhang et al., 2024a), and hence, Lemma F.12 holds.

**Lemma F.12** (c.f. Lemma 3.5 of Zhang et al. (2024a)). *Balance is optimal on $\mathcal{G}$.*

Together with Lemma F.11 and Lemma F.12, it holds that no algorithm does better than $0.594$ in OMSR on $\mathcal{G}$.

