# OpenReview forum: "DiMa: Understanding the Hardness of Online Matching Problems via Diffusion Models"
_ICML.cc/2025/Conference — ICML 2025 poster_

### Official Review · Reviewer_uC52 · 2025-02-27

**Overall Recommendation:** 3

**Summary:**

The authors study the hardness of the online bipartite matching (OBM) problem using denoising diffusion probabilistic models (DDPMs). The DDPMs are trained using policy gradient to generate hard instances for OBM. For classic OBM problem this represents the hardest input showing the validity of this  approach. Finally, this method is used to improve the hardness upper bounds for some variants of OBMs.
In particular, with random arrivals it improves the bound from 0.727 to 0.723, and with stochastic arrivals it improves the bound from 0.597 to 0.594.

**Claims And Evidence:**

DDPM with specific fine tuning can help with generating hard instances in variants of the OBM problem.  For OBM with random arrivals it improves the bound from 0.727 to 0.723, and with stochastic arrivals it improves the bound from 0.597 to 0.594.

This is proved by training DDPMs, and then proving the hardness theoretically on specific instances.

**Essential References Not Discussed:**

I am not familiar with the literature.

**Ethical Review Concerns:**

Theoretical paper. Hard to foresee impact.

**Experimental Designs Or Analyses:**

The experiment design seems reasonable given the objective of finding hard instances.

**Methods And Evaluation Criteria:**

The method  - training DDPM to find hard instances, and showing hardness of these instances - seems reasonable.

**Other Comments Or Suggestions:**

N/A

**Other Strengths And Weaknesses:**

Strengths: The authors develop a new AI based techniques to produce hard instances. This seems to improve the state-of-the-art for two variants of the OBM problems.

Weakness:
- The resulting hard instances look very similar to the existing ones. Thus, although the methodology seems novel the results are probably underwhelming. Specifically, given the similarity simply tuning a few parameter could have given us the exact result.
- It seems we may not be able to discover novel settings by only mimicking the existing hard instances. I feel AI can be helpful only if we can somehow combine our intuitions (e.g. fine-tune with known hard instance) with novel exploration strategies. Without exploration we are bound to converge to known results.
- I could not find a proper mechanism to extrapolate the patterns found from finite sized instances to general instances (with n vertices, for any n). Why not train DDPMs to generate rules that can be extrapolated?

Edit after rebuttal: I read the clarifications provided by the authors. I appreciate the direction of the work even though I still maintain some of the reservations. I will keep my score.

**Questions For Authors:**

- Why the produced hard instances are close to the existing ones? Were they used for fine tuning? It seems exploration is lacking here.

- Can we prove the optimality of the existing algorithms on novel instances, even if they are found? If not then it feels there is a gap in the current approach.

**Relation To Broader Scientific Literature:**

The application of AI to design hard instances to understand complexity is of interest to theoretical computer science community. However, I am not qualified to comment on it's importance.

**Theoretical Claims:**

The theoretical claims make sense at a high level. I have not checked the details of the proof.

---

> ### Author Rebuttal · Authors · 2025-04-01
>
> We appreciate your constructive review and support of our work in applying AI techniques to theoretical computer science. This encourages us to study further in this direction. We hope the following responses will address your concerns and look forward to your ongoing support:
>
> W1: While apparently similar, we believe the new hard instances fundamentally differ from existing ones and are non-trivial to identify because: For example, in the OMRA problem as demonstrated in Figure 5(a) and Figure 6, hard instances we find have an additional dense subgraphs in the bottom left corner, which exhibit distinct characteristics compared to known distributions. Theorem 5.1 and the Appendix provide rigorous theoretical proofs confirming they get a lower upper bound. Conventional hard instance construction relies heavily on expert-designed structures (e.g., thick-z to upper-triangular configurations in OBM problems), leaving a large search space of $2^{n/2}$ intermediate graphs. Our methodology overcomes this limitation through the combination of diffusion models and RL. By directly optimizing objective functions (competitive ratio) through reward-guided exploration, we have improved the efficiency of exploration while enabling the discovery of harder instances.
>
> W2: We acknowledge that the exploration is necessary in discovery. Our DiMa actually employs an innovative exploration strategy. The exploration occurs during the fine-tuning process using RL. DiMa consists of two phases. The first phase pretrains a DDPM, aiming to learn some known hard distributions. Much more importantly, DDPM is known to be effective in generating **diverse** samples, which serves as an exploration. Subsequently, the RL fine-tuning process optimizes the sample generation by distilling those harder instances.
>
> W3: Thank you for proposing the idea related to rule generation. This is actually a very interesting but till now still very challenging topic that we plan to leave as future work. In theoretical computer science, generalizing fixed graph size, say, $n$, to arbitrary graph sizes are roughly crafted by human expertise. However, very large instances are mainly constructed by repeating the small hard instance structures.  Therefore, to facilitate rule generation, the fundamental step is to search and identify these hard structures (as what we did in this paper), which were previously hand-crafted based on expert insights. Our contribution is to propose a novel AI-assisted framework to achieve that goal. Nevertheless, we acknowledge that it is still a preliminary attempt in the direction of AI for TCS; We hope we figure out how to generate rules instead in the future.
>
> Q1: As mentioned in the response for W1, the hard instances we constructed are not close to the existing ones. Additionally, these known hard instances are not used for fine-tuning. In fact, the known hard instances are used in the pretraining process of DiMa to train a DDPM, allowing it to learn the known hard distributions and generate diverse instances.
>
> Q2: Yes, we prove the optimality of the existing algorithms on novel instances which is shown in Appendix F. More details is as follows:
>
> - For OMSR, we prove in Appendix F.2 that Balance (the state-of-the-art algorithm) is optimal on the instances generated by DiMa.
> - For OMRA, while no algorithm is proven universally optimal, our instances theoretically establish that Ranking’s competitive ratio degrades to 0.723—a new upper bound which is proved in Appendix F.1.
>
> This ensures that the generated instances are not merely harder empirically but also theoretically valid.

---

> > ### Comment · Reviewer_uC52 · 2025-04-02
> >
> > I thank the authors for their thoughtful comments. I agree that this direction of AI assisted TCS is exciting. However, I still have the following reservations
> > - relying on the inherent diversity of DDPM may not be effective, and we may end up at the local neighborhood of the hard instances used to pre-train the model.
> > - the requirement to show optimality of existing algorithms on novel instances keeps a reasonable burden out of AI's reach (can we combine reasoning models? requires some validation).
> >
> > I will maintain my score.

---

> > > ### Author Response · Authors · 2025-04-09
> > >
> > > Great thanks for your kind acknowledgment and response. We hope the following response can address your concerns:
> > >
> > > Q1: While the local optimality happens in nearly all AI-driven applications, our framework makes every effort to avoid such situations, such as training an RL-guided DDPM for generation or introducing appropriate randomnesses in the implementation. We clarify that our DiMa rarely relies on the initialized instances for DDPM pre-training, meaning that DiMa still works even if it starts with some random instances. See the rebuttal to Reviewer ibv8 in the 'Generalizability' paragraph for details and additional experiments. Further, in our first application, say, the classic online bipartite matching model, DiMa succeeds in converging at the very global optimum, i.e., the upper triangular instances, indicating the great potential of DiMa's capability in finding entirely novel instances. For the two open problems, we can not verify the optimality of our constructed instances, essentially because only when we know what the exact optimal algorithms are, meaning that the algorithmic lower bound meets the hardness upper bound, we could say that some hard instances reach the worst. However, this is another super challenging online matching theory task of independent interest. Nevertheless, even if they still fell into the local neighborhood, knowing the worse could be always on the way of knowing the worst.
> > >
> > > Q2: We actually see the potential of reasoning models in our task, while it is kind of a mystery based on the current research. To the best of our knowledge, in very recent advances, reasoning models may succeed in some traditional mathematical proofs with some explicit sequential logics, after they see most of the standard proofs. However, in TCS, most tasks can be much more complex. While proving a bunch of lemmas is significant, verifying what to prove can be much more tricky and challenging. It relies on such as how to model the problem, how to write some linear or non-linear programs, or any other diverse mathematical tricks. In our motivation of hardness understanding, our method is to separate the hard instances construction as an independent subtask, which we surprisingly find and validate that AI techniques can help. However, in a broader scope of understanding online optimizations, to the best of our knowledge, how to benefit from AI gains little evidence. We thank the reviewer for the insight in mentioning reasoning models, and we plan to leave the attempt of reasoning models in online algorithms as one of the most valuable future directions for us. However, we tend to believe this is an independent story from the contributions we made in this paper.
> > >
> > > We thank you again for your great efforts in reviewing our paper and really appreciate your valuable insights and discussions.

---

### Official Review · Reviewer_MM8W · 2025-03-09

**Overall Recommendation:** 4

**Summary:**

The paper presents a method based on a diffusion model trained using reinforcement learning. This model is then used to generate difficult instances for specific algorithms in online bipartite matching problems. The method successfully generates hard examples in two variants of the online bipartite matching problem, leading to an improved upper bound on the competitive ratio for these problems.

**Claims And Evidence:**

Yes

**Essential References Not Discussed:**

Not that I am aware of.

**Experimental Designs Or Analyses:**

Yes, I checked all the details in the main text.

**Methods And Evaluation Criteria:**

Yes

**Other Comments Or Suggestions:**

The authors devote a significant portion of the paper to explaining elementary details on diffusion models and RL fine-tuning. I believe this discussion could be condensed in the main text, with the details deferred to the Appendix. Instead, I would prefer to see a more in-depth discussion of the hard instances identified by their model in the main text.

**Other Strengths And Weaknesses:**

This is a strong paper. It is impressive that generative models can be used to advance state-of-the-art theoretical results. Since the paper delivers on its promise, I did not find any obvious weaknesses.

**Questions For Authors:**

For the instances that improved the state of the art, how many samples were generated before discovering the hard instance? Was it among the points in the $100$ trajectories generated during fine-tuning?

**Relation To Broader Scientific Literature:**

This paper improves the state-of-the-art by improving the upper bounds on the competitive ratio for two well-known variants of the online bipartite matching problem. Moreover, it is a valuable contribution to the development of applied methods that can improve theoretical analysis.

**Theoretical Claims:**

The proofs are in the appendix, so I did not check the details.

---

> ### Author Rebuttal · Authors · 2025-04-01
>
> We sincerely appreciate your constructive feedback and acknowledgment that "it is a valuable contribution to the development of applied methods that can improve theoretical analysis." We hope the following responses address your concerns and look forward to your ongoing support:
>
> Comments or Suggestions: Before submitting our manuscript, we indeed faced a dilemma regarding the arrangement of content due to space constraints—whether to focus on the presentation of the AI-based framework or to delve into the theoretical analysis of the newly constructed hard instances. Ultimately, we chose the former based on the following considerations: (1) Given that the submission targets an international machine learning conference, we strive to enable researchers in the field to easily grasp the essence of our approach and appreciate the connection between our AI-based method and the theoretical domain; (2) Beyond the theoretical advancement, one of the major contributions of our work is the novel shortcut policy gradient (SPG) optimization method. We aim to present the essential details on diffusion models and RL fine-tuning in the preliminaries section to ensure that readers fully comprehend how SPG integrates into the framework. We greatly appreciate your encouragement to reconsider the content arrangement of our manuscript. In response, we have revised the manuscript by condensing the background explanations and moving them to the appendix while increasing the analysis of the newly constructed hard instances in the main text.
>
> Q1: We claim that the hard instances were indeed generated during fine-tuning among the points in the 100 trajectories. In our framework, DiMa typically converges within 100 epochs and successfully identifies harder instances within the 10000 samples generated (100 trajectories × 100 epochs). During RL fine-tuning, we sample 100 trajectories per epoch to update the policy. This design stems from the inherent requirement of RL training, where gradient updates rely on accurately estimating the expected reward from a sufficiently large set of trajectories. Using fewer samples would result in a high variance in gradient estimation, leading to unstable policy updates. Through empirical validation, we found that sampling approximately 100 trajectories per epoch strikes a balance between stable policy convergence and computational efficiency. As the training progresses, the generated instances gradually converge toward harder candidates, and the final hard instances naturally emerge from the sampled trajectories in later epochs.

---

> > ### Comment · Reviewer_MM8W · 2025-04-01
> >
> > I thank the authors for their response. I still think that there are too many elementary details in the paper's main text and not enough details of the actually interesting part. However, it is ultimately up to the authors to decide the best way to present their work. In light of their response, I would like to keep my original review and score.

---

> > > ### Author Response · Authors · 2025-04-08
> > >
> > > Thanks again for your kind review and valuable suggestion. We are continuing to polish our paper for better presentation.

---

### Official Review · Reviewer_ibv8 · 2025-03-14

**Overall Recommendation:** 3

**Summary:**

The paper introduces a novel framework called DiMa, which enhances the theoretical understanding of Online Bipartite Matching (OBM) problems using diffusion models. DiMa models the generation of hard instances as a denoising process and optimizes them using a new reinforcement learning algorithm called Shortcut Policy Gradient (SPG). The framework is examined on the classic OBM problem, where it successfully reproduces the known hardest input instance. DiMa is also applied to two open-ended OBM variants, improving their theoretical state-of-the-art upper bounds by indentifying worst cases with even lower rewards.

**Claims And Evidence:**

The paper provides promising evidence for the effectiveness of DiMa in improving the understanding of OBM problems. The results on reproducing known hard instances and improving bounds on OBM variants are compelling. However, further clarification and stronger evidence in areas such as hyperparameter sensitivity, generalizability, complexity, and a deeper analysis of the generated hard instances would make the claims even more convincing.

*  The paper provides evidence that DiMa can reproduce the known hardest instance for the classic OBM problem, even without seeing such instances in the training set. For instance, it can discover triangular graph during the RL tuning stage, even if it's only trained with thick-z graph. This is a very important property, as it demonstrates the framework's ability to learn and generate known hard instances. For instance

* The paper presents results showing improved upper bounds for two OBM variants (online matching with random arrivals and online matching with stochastic rewards). Compared with the ICML 2024 paper, (Zhang et. al.), the instances found by DiMa outperforms this baseline in several different graph sizes. The improved bounds are supported by specific instances generated by DiMa, and the authors provide proof sketches in the appendix.

***Weakness***

* While the paper highlights the significance of selecting distribution q and tuning hyperparameters, the methodology for determining an 'appropriate' q and the sensitivity analysis of hyperparameters require further elaboration. The authors' use of thick-z, though empirically successful, necessitates expertise and may limit generalizability. This reliance on a specific distribution also potentially undermines the paper's goal of reducing the expertise required compared to the ICML baseline. A more detailed examination of the impact of these choices on the results and the method's robustness would be valuable.

* The computational complexity of DiMa, especially concerning the training of DDPMs and the fine-tuning process with RL, could be discussed in more detail.  While the authors address some computational concerns with SPG, a thorough analysis of the method's scalability to larger problem instances would be valuable.

* While the paper claims to generate "novel" hard instances, it would be interesting to see a more in-depth comparison of the generated instances with existing ones.  A more rigorous analysis of the differences and the specific properties that make the generated instances harder would further support this claim.

**Essential References Not Discussed:**

Given this paper's focus on online bipartite matching (OBM) using machine learning, particularly within the 'ML for OBM' section of related works, including the following works on RL-based OBM algorithms would be beneficial.

* Alomrani, Mohammad Ali, Reza Moravej, and Elias B. Khalil. "Deep policies for online bipartite matching: A reinforcement learning approach." arXiv preprint arXiv:2109.10380 (2021).
* Li, Pengfei, Jianyi Yang, and Shaolei Ren. "Learning for edge-weighted online bipartite matching with robustness guarantees." International Conference on Machine Learning. PMLR, 2023.

**Experimental Designs Or Analyses:**

The experimental designs and analyses presented in the paper are generally sound and provide evidence for the effectiveness of DiMa. In their ablation studies, they also analyze the impact of parameter $\gamma$ or distribution q on the overall performance. The comparison of the proposed SPG algorithm with a baseline method (DDPO) provides evidence for its effectiveness by skipping some steps.

However, there are areas where further details, analysis, and more rigorous evaluation would enhance the validity and strengthen the conclusions. Addressing the points mentioned above, such as providing a more thorough hyperparameter analysis, justifying the choice of q (especially in more difficult senarios), and including generalizability analysis, would contribute to a more convincing and robust evaluation.

**Methods And Evaluation Criteria:**

***Proposed methods***
* Using Denoising Diffusion Probabilistic Models (DDPMs) to generate hard instances for Online Bipartite Matching (OBM) is a novel approach.  DDPMs are known for their ability to generate high-quality samples while capturing the underlying distribution of the training data.  In this context, it's reasonable to use them to generate challenging OBM instances by learning from a distribution of known hard cases.

* Fine-tuning the DDPM using reinforcement learning (RL) is also a sensible choice.  RL allows the model to optimize the instance generation process based on a reward signal that reflects the hardness of the generated instances.  This makes it possible to iteratively refine the instance distribution towards generating harder and harder cases.

* The proposed Shortcut Policy Gradient (SPG) algorithm addresses a key challenge in applying RL to this problem. The experimental results suggest that SPG outperforms other methods like DDPO, but such advantage is mainly brought by DDIM instead of a novel RL algorithm.

***Metrics***
* The competitive ratio (CR) is a standard metric for evaluating the performance of online algorithms, as well as OBM problems.  It measures the ratio between the matching size found by an online algorithm and the optimal offline matching.
* The goal of the paper is to improve the theoretical upper bounds on the competitive ratio for OBM problems. Ideally, to evaluate the hardness of some OBM problems, the upper bound should be defined in terms of the problem instead of algorithms. But in some empirical studies (Sec 5.1 & 5.2), it seems like these CRs are evaluated with respect to some algorithms. But I think it's acceptable in some existing works.

In summary, the proposed methods and evaluation criteria are well-aligned with the goal of understanding and improving the hardness results for OBM problems.

**Other Comments Or Suggestions:**

N.A

**Other Strengths And Weaknesses:**

N.A

**Questions For Authors:**

N.A

**Relation To Broader Scientific Literature:**

* The paper builds upon the emerging field of AI-enhanced combinatorial optimization, especially the online bipartite matching problems. It specifically acknowledges the work of Zhang et al. (2024a)  as a recent attempt in this area, which uses reinforcement learning to improve the hardness result of an OBM model.  DiMa contributes to this area by introducing a novel framework based on diffusion models (DDIM and DDPM).

* The use of denoising diffusion probabilistic models (DDPMs) connects the paper to the broader literature on generative models.  DDPMs have been successful in various applications, particularly in image generation, and DiMa leverages their generative capabilities to create hard instances for OBM problems. The paper also draws inspiration from denoising diffusion implicit models (DDIM), which are known for efficient sampling.

* The fine-tuning process in DiMa utilizes reinforcement learning (RL), which connects the work to the extensive literature on RL algorithms and their applications. The paper also proposes a novel RL algorithm, Shortcut Policy Gradient (SPG), which builds upon existing policy gradient methods like REINFORCE

**Theoretical Claims:**

Upon reviewing Theorems 5.1 and 5.2, which assert a superior competitive ratio bound compared to prior research, the proof's logical progression appears sound. I presume the numerical results, derived from their equations, are accurate.

---

> ### Author Rebuttal · Authors · 2025-04-01
>
> Thank you for your comprehensive and insightful reviews and for positively evaluating our paper and contributions as “promising, compelling, and important”. We hope the following clarifications will address your concerns and look forward to your reconsideration of our work. Our supplemental results are in supplemental figures:https://anonymous.4open.science/api/repo/rebuttal_figure/file/Reviewer%20ibv8.pdf?v=8c96977d.
>
> Hyperparameter sensitivity: Besides Figure 11 in the Appendix, we have added a more detailed sensitivity analysis of $\gamma$ in the ablation study (i.e., $\gamma$ varying from 0 to 1 at 0.05 intervals). Our DiMa easily converges to the worst upper-triangular instances within 100 epochs when $\gamma \in [0.2, 0.35]$, while for small ($\gamma < 0.10$) or large ($\gamma > 0.50$) values, it fails to converge even after 500 epochs. See Figure 1 in supplemental figures.
>
> Generalizability: We think there might be a misunderstanding in DDPM initialization. We clarify that DiMa can converge to upper triangulars starting with diverse distributions (not restricted to the thick-z), even with random samples. Our main argument is that leveraging structural information from known hard instances, which is easily obtained in literature, may facilitate the fine-tuning process to discover new harder instances. Such an idea also aligns with conventional hand-crafted constructions in OBM, where new harder instances are typically built on the existing ones with some slight modifications. Nevertheless, we thank you for pointing out this potential ambiguity, and we have added detailed explanations in the experimental setup. We also provide the following evidence to further support them: (1) We visualize some of initialization distributions that successfully converge to upper triangulars, including randomly sampled ones (Figure 2 in supplemental figures); (2) We conduct additional experiments starting with 50 random samples, nearly one thirds of which enables DiMa to find the worst instances within 100 epochs. Finally, we emphasize that unlike [Zhang et al.], which appears to heavily depend on intermediate observations for iterative adjustments during the training process to discover novel instances, our method largely reduces reliance on expert insights because existing hard instances are easily obtained in literature. Although DiMa can benefit from known hard instances, it is essentially end-to-end.
>
> Complexity: Thank you for your acknowledgment of the effectiveness of our SPG in reducing computational costs. We propose SPG as an independent contribution addressing a common challenge in the ML-for-OM literature. Traditional methods struggle with large-scale graphs. For example, in [Zhang et al.], their RL approach for OMSR seems limited to small graphs (less than 10 offline vertices). In contrast, DiMa efficiently works on remarkably larger instances (of size larger than 50) on a 24GB GPU within an hour. More experimental costs of DiMa across various graph sizes are presented in the table below:
>
> | Method                  | Graph size | Memory size | Running time |
> | :---------------------- | :--------: | :---------: | :----------: |
> | SPG                     |   12x12    |   0.62GB    |  20s/epoch   |
> | SPG                     |   20x20    |   1.14GB    |  25s/epoch   |
> | SPG                     |   40x40    |   5.05GB    |  35s/epoch   |
> | SPG                     |   80x80    |   22.59GB   |  45s/epoch   |
> | Traditional Computation |   20x20    |    >24GB    |      --      |
>
> We thank the reviewer for pointing this out and will add a separate discussion to highlight such strength of our DiMa.
>
> Analysis of hard instances: We do identify properties distinct from those reported before. For example, in OMSR, our baseline of [Zhang et al.] claims they observe two primary properties named Consistency and Exclusivity property. We actually discover a more fine-grained property critical for harder instances, which is what we call the significance of dense subgraphs. While the ICML baseline seems to implicitly suggest that including some dense structures while ensuring sparsity in other parts is helpful for hardness construction, we further observe the impact of the proportion and location of dense subgraphs in generating harder instances. Similar findings also arise in OMRA. The proofs of Theorems 5.1 and 5.2 are built on these findings distilled from learned instances (in Appendix F). We appreciate the reviewer's suggestion and have added a more detailed discussion in the 'proof sketch'. Additionally, we have added an extra discussion at the end of Section 5 to clarify the differences between our property and the baseline.
>
> Related work: Thank you for providing valuable literature. We have added these two papers to the related work section ('ML for OBM' paragraph).

---

> > ### Comment · Reviewer_ibv8 · 2025-04-02
> >
> > I really appreciate the author's rebuttal and efforts in additional experiments. I think most of my concerns are addressed and I will raise my evaluation accordingly.

---

> > > ### Author Response · Authors · 2025-04-08
> > >
> > > Thanks again for your thorough review and the acknowledgment of our rebuttal.

---

### Official Review · Reviewer_zCLi · 2025-03-15

**Overall Recommendation:** 3

**Summary:**

In this paper, they train a diffusion model to construct hard instances for Online Bipartite Matching problem.
Using the proposed method they find state-of-the-art upper bounds for  the random arrivals and stochastic arrivals variants of Online Bipartite Matching problem.
For the training of the diffusion model they propose a reinforcement learning technique named shortcut policy gradient.

**Claims And Evidence:**

The main claim of the paper is the construction of hard instances for two variants of Online Bipartite Matching.
Apparently, this claim is clear and has convincing evidence to support it.

What I am not sure about is the usefulness and generalizability of the proposed method.
It seems that it finds instances very close to already known hard instances.
Hence, it is not clear whether this can enable any progress in the theoretical understanding of the problems.
Also, it is not clear how this method compares to other intuitive methods that one might try in order to find hard instances.
For example, it seems possible that applying MCTS (or some other search technique) we might also find harder instances than the known hard instance. (see also the Questions section)

**Essential References Not Discussed:**

N/A

**Experimental Designs Or Analyses:**

I didn't find any issue in the experiments.

**Methods And Evaluation Criteria:**

The proposed methods and evaluation criteria make sense.

**Other Comments Or Suggestions:**

Some minor errors/typos:
- Move Algorithm 1 to the top of the page
- line 260:  "at the neighbor" -> "at the neighborhood"
- line 174: "we without loss of generalization assume" needs rephrasing

**Other Strengths And Weaknesses:**

I think some points need further clarification. See the Questions section.

**Questions For Authors:**

- In equation (6), you state that $J_{RL}(\theta) \propto J_{CR}(\theta)$. However, $J_{RL}(\theta)$ only takes into account the reward of the algorithm and not the reward of the optimal solution. Hence, are these two quantities proportional?

- Did you try applying the proposed technique to other problems as well? Will applying your method to other problems require any modifications?

- Can you explain how the formulatation of the reverse denoising process MDP (section 4.2)?

- Is there any particular reason you used the rounding in equation (9). For example, you set $\hat{I}_{ij}$ probabilistically and not according to a predetermined threshold.

- Do you have any understanding about the hard instances your model found? Do you think they can be used to make progress in the theoretical study of the problem?

- What will happen if you train your model based on the new hard instance you constructed? Do you expect it to find an even harder instance? Does this approach make sense? Please explain.

**Relation To Broader Scientific Literature:**

The main related work in the domain of AI-enchanced combinatorial optimization theory is that of Zhang et al. (2024a).
This paper provides a more structured approach for the problem of interest by improving various parts of the previous approach.

In the domain of online combinatorial optimization and specifically online bipartite matching this paper provides state-of-the-art upper bounds for two variants of the problem.

**Theoretical Claims:**

N/A. The paper is mostly experimental.

---

> ### Author Rebuttal · Authors · 2025-04-01
>
> We sincerely appreciate your thorough review and positive review of our work as "clear and convincing". We hope the followings address your concerns and look forward to your kind reconsideration of our work:
>
> Generalizability(also Q2): We believe our DiMa demonstrates strong generalizability. We evaluate DiMa on **three** different OBM problems, among which, two **theoretical SOTAs** are improved (See Appendix F for formal proofs). In particular, the upper bound of OMRA has not been improved since 2011. Similar previous works (such as [Zhang et al.] or [Kong et al.]) can either work on one specific problem or just reproduce (not improve) the known results. Further, our DiMa can be easily adapted to other OBM problems with slightly different implementations, including initializations and hyper-parameters. Finally, though the learned hard instances look similar to the known ones (they are actually not quite similar), conventional search methods like MCTS may behave like brute forces, which are super inefficient in finding worse instances. In contrast, our RL-guided DDPM framework generates low-CR instances through objective-driven rewards, enabling efficient targeted exploration toward harder instances. To further validate the generalizability of DiMa, we evaluate it on another famous AdWords problem, which will be included in the Appendix. See supplemental figures: https://anonymous.4open.science/api/repo/rebuttal_figure/file/Reviewer%20zCLi.pdf?v=d722f741 for details.
>
> Q1: We thank the reviewer's insight on the potential ambiguity of reward calculation. CR requires comparing algorithmic solutions with offline optimum (OPT). In our paper, to simplify computation, we constrain all instances to have a perfect one-to-one matching in OPT, such that the OPT always equals the number of offline vertices. Such treatment aligns with nearly all (theoretical or AI-driven) prior works. To mitigate this potential ambiguity, we have added detailed explanations of CR to Section 3.1.
>
> Q3: The MDP of the reverse denoising process is:
>
> - State: Represent as a tuple $(\mathcal{I}_{T-t},T-t)$ (current noised instance and timestep index). This fully captures the system state without historical information, satisfying the Markov property.
>
> - Action: The role of the policy $\pi_\theta$ at each step $t$ is to denoise the current noised instance, and the resulting denoised instance $\mathcal{I}_{T-t-1}$ is the action we take.
> - Reward: Rewards are focused on the quality of $\mathcal{I}_0$, with no rewards for intermediate steps.
>
> - Transition: The next state $ s_{t+1} = (\mathcal{I} _ {T-t-1}, T-t-1 )$ can be uniquely determined by $P(s_{t+1}|s_t,a_t)$.
>
> - Policy: The denoising network $p _ θ(\mathcal{I} _ {T-t-1}|\mathcal{I} _ {T-t})$ determines how to generate the next denoised instance, which is the specific implementation of the policy.
>
> We have updated the above to Section 4.2.
>
> Q4: We actually tried both probabilistic sampling and predetermined thresholding for rounding. While both methods can work, the thresholding method is much more stable and easier to implement. In contrast, probabilistic sampling results in higher variance in the reward distribution, leading to increased bias in policy gradient estimation and a reduction in the effective step size of policy updates.
>
> Q5: The hard instances found by DiMa correspond to **theoretical** upper bounds (with formal proofs in Appendix F) of both OMSR and OMRA, enhancing the theoretical understanding of these two open problems. Our proofs benefit from some structural properties observed and distilled from the learned instances. For example, in OMSR, we identify a fine-grained property named the significance of dense subgraphs. We observe that it is crucial to include some dense structures while ensuring sparsity in other parts. Further, we also see the impact of the proportion of dense subgraphs in obtaining harder instances. The proof of Theorem 5.2 is built on these findings distilled from learned instances.
>
> Q6: We thank the interesting question. We actually once tried to continuously train DiMa on the new hard instances we found. However, it fails to produce any harder ones. We expect we have obtained the hardest instances, even if we can not verify them now unless the optimal algorithms are found. Nevertheless, we acknowledge that this is a valuable question that can further support the effectiveness of our DiMa, and we have included it as a separate discussion.
>
> [Zhang et al.] :  Zhang Q, Shen A, Zhang B, et al. Online matching with stochastic rewards: provable better bound via adversarial reinforcement learning. ICML 2024.
>
> [Kong et al.] : Kong W, Liaw C, Mehta A, et al. A new dog learns old tricks: RL finds classic optimization algorithms. ICLR 2018.

---

### Decision · Program_Chairs · 2025-05-01

**Decision:**

Accept (poster)

**Comment:**

This paper uses diffusion models with RL-inspired tuning that is able to generate hard instances for combinatoric problems.
All reviewers believe the methodology to be sound. It is a purely empirical paper, so there are no critical proofs to check.

They successfully generate new problem instances which provide improved worst-case bounds for two variants or OBM, which is clear empirical evidence that their method is useful.